# ScaleKD: Strong Vision Transformers Could Be Excellent Teachers

**Jiawei Fan**[*]
Intel Labs China
`jiawei.fan@intel.com`

**Chao Li**[*]
Intel Labs China
`chao3.li@intel.com`

**Xiaolong Liu**[*]
iMotion Automotive Technology
`xiaolong.liu@imotion.ai`

**Anbang Yao**[*][†]
Intel Labs China
`anbang.yao@intel.com`

## Abstract

In this paper, we question if well pre-trained vision transformer (ViT) models could be used as teachers that exhibit scalable properties to advance cross architecture knowledge distillation research, in the context of adopting mainstream large-scale visual recognition datasets for evaluation. To make this possible, our analysis underlines the importance of seeking effective strategies to align (1) feature computing paradigm differences, (2) model scale differences, and (3) knowledge density differences. By combining three closely coupled components namely *cross attention projector*, *dual-view feature mimicking* and *teacher parameter perception* tailored to address the alignment problems stated above, we present a simple and effective knowledge distillation method, called *ScaleKD*. Our method can train student backbones that span across a variety of convolutional neural network (CNN), multi-layer perceptron (MLP), and ViT architectures on image classification datasets, achieving state-of-the-art knowledge distillation performance. For instance, taking a well pre-trained Swin-L as the teacher model, our method gets 75.15%|82.03%|84.16%|78.63%|81.96%|83.93%|83.80%|85.53% top-1 accuracies for MobileNet-V1|ResNet-50|ConvNeXt-T|Mixer-S/16|Mixer-B/16|ViT-S/16|Swin-T|ViT-B/16 models trained on ImageNet-1K dataset from scratch, showing 3.05%|3.39%|2.02%|4.61%|5.52%|4.03%|2.62%|3.73% absolute gains to the individually trained counterparts. Intriguingly, when scaling up the size of teacher models or their pre-training datasets, our method showcases the desired scalable properties, bringing increasingly larger gains to student models. We also empirically show that the student backbones trained by our method transfer well on downstream MS-COCO and ADE20K datasets. More importantly, our method could be used as a more efficient alternative to the time-intensive pre-training paradigm for any target student model on large-scale datasets if a strong pre-trained ViT is available, reducing the amount of viewed training samples up to 195×. The code is available at `https://github.com/deep-optimization/ScaleKD`.

## 1 Introduction

**Background.** The great success of deep learning in computer vision (CV) has been driven by an explosion of neural network architectures among which convolutional neural networks (CNNs) [1–3], vision transformers (ViTs) [4, 5] and multi-layer perceptrons (MLPs) [6–8] are three major model categories. While CNNs were the de facto models for about a decade, recent progress shows that large ViT models have attained state-of-the-art performance on many visual recognition tasks such as image

---

[*] Core authors contributed to method formulation, experimental design and analysis.

[†] Corresponding author.

38th Conference on Neural Information Processing Systems (NeurIPS 2024).

classification, image segmentation, and object detection. In principle, ViTs extend the philosophy of predominant transformer architectures [9] in natural language processing (NLP) to vision tasks. They convert an image into a sequence of equal-sized patches treated as tokens resembling words in NLP, then apply the dot-product self-attention mechanism over the sequence of image patches. ViTs designed in this way couple with a powerful data-hungry learning paradigm: models are first pre-trained on massive datasets (with supervised or self-supervised [10, 11] or cross-modality learning [12, 13]) and then fine-tuned on target datasets (with supervised learning). As the size of ViT models or pre-training datasets increases, the pre-trained models tend to have improved generalization performance. Despite this notable model performance scalability, the pre-training process of ViTs leads to significantly huge expenses. Furthermore, large pre-trained ViTs are memory-hungry and computationally intensive, prohibiting their deployment in many resource-constrained application scenarios. In contrast, CNNs and MLPs are still widely used in industry, due to the wider availability of effective implementations and optimizations compared to ViTs.

**Motivation of This Work.** In parallel, knowledge distillation (KD) has proven to be a promising model compression pathway and has attracted lots of research interests. It relies on a teacher-student framework that transfers the knowledge learned by a large teacher model to a compact student model, aiming to make the student model can have improved performance to substitute the teacher model in deployment. However, most existing KD methods [14–35] focus on CNN architectures, and usually perform evaluation on small datasets with non-mainstream student models for industrial applications, lagging far behind the evolution of neural network architectures. Although there have been few recent efforts [36–39] on using ViT teachers, they explore narrow focuses that use small ViT teachers without pre-training on massive datasets, following the ways previously studied in CNN-based KD methods. In this paper, *we attempt to connect knowledge distillation research with well pre-trained ViT models that stand out for their remarkable scalability, via a new viewpoint*. Specifically, *we question* whether well pre-trained ViT models could be used as teachers that effectively transfer their scalable properties to target student models having different typed architectures such as CNN and MLP or heterogeneous ViT structures (we refer 'cross architecture KD' to such a more generalized formulation in this work), in the context of using mainstream large-scale visual recognition benchmarks.

**Problem Analysis.** To answer the question in our motivation, we think the knowledge transfer difficulties are rooted in the following three aspects of differences: (1) *Differences in feature computing paradigm*. In terms of semantic units, ViTs operate on a sequence of equal-sized image patches added with positional embeddings, whereas CNNs operate on regular grids of pixels. In terms of core operations, ViTs rely on self-attention operations to model global feature dependencies, whereas CNNs rely on convolution operations to model local features. Although MLPs also use a patchify stem as ViTs, they rely on fully connected operations instead of self-attention operations and do not use positional embeddings, showing inferior feature learning ability. These differences in feature computing paradigm pose the first knowledge transfer barrier to overcome. (2) *Differences in model scale*. On the micro scale, model scale differences among ViTs, CNNs, and MLPs lie in network width, network depth, building blocks, etc. On the macro scale, model scale differences come from the capability of scaling the model size for ViTs, CNNs and MLPs towards better performance and generalization ability. As a result, these differences in model scale make the capacity of different network architectures typically vary significantly, emerging as the second knowledge transfer barrier to address. (3) *Differences in knowledge density*. Under the prevalent pre-training and fine-tuning paradigm, when scaling up pre-training datasets, large ViTs usually exhibit obviously superior performance scalability than top-performing CNNs and MLPs in terms of fine-tuning on both upstream image classification tasks and downstream dense prediction tasks [40, 41]. As for knowledge distillation in this work, we assume that pre-training datasets are no longer accessible and only well pre-trained ViT teacher models are available, avoiding the expensive pre-training process and making the setting well suited for real applications. Under this context, when training student models on upstream image classification datasets like ImageNet-1K, the knowledge density between teacher and student models is different, which appears as the third barrier to handle. From the above analysis, we can conclude that the design of effective schemes to align (1) feature computing paradigm differences, (2) model scale differences, and (3) knowledge density differences between the pre-trained ViT teacher and target student models, plays the key role to attain our goal.

**Design Insights and Contributions.** Accordingly, we present Scalable Knowledge Distillation (ScaleKD), a simple and effective cross architecture KD method, which addresses the above difficulties in a progressive manner. Fundamentally, to bridge the feature computing paradigm differences

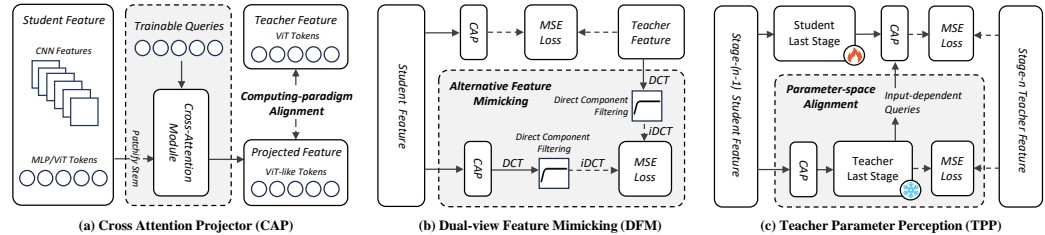

**(a) Cross Attention Projector (CAP)**      **(b) Dual-view Feature Mimicking (DFM)**      **(c) Teacher Parameter Perception (TPP)**

Figure 1: Overview of three core components in our ScaleKD, which are (a) cross attention projector, (b) dual-view feature mimicking, and (c) teacher parameter perception. Note that the teacher model is frozen in the distillation process and there is no modification to the student's model at inference.

between ViT and the other heterogeneous architectures, we propose *cross attention projector* (CAP, shown in Figure 1(a)), motivated by some previous works [42–44] that utilize cross attention mechanisms to align different modalities. For semantic unit differences, CAP utilizes positional embeddings and a patchify stem to transform the semantic units of CNN and MLP into transformer-like tokens. To further bridge core operation differences, CAP employs cross-attention operation and trainable queries that share the same attributes as the teacher's features to model global interdependencies on the student's features. In this way, CAP could align computing paradigm differences between the ViT teacher and the heterogeneous student in form, serving as the base component in our method.

Different from feature computing paradigm differences, model scale differences and knowledge density differences are not explicitly and separately modeled in the KD process, as they are intertwined under the prevailing pre-training and fine-tuning paradigm and are finally encoded in teacher and student models' feature space and parameter space. In light of this, we investigate both feature and parameter spaces of teacher and student models and observe two critical phenomena:

- **Feature Space:** As shown in Figure 2 and Figure 5, the frequency distributions of the features for the pre-trained ViTs are extremely imbalanced, where the direct component (zero frequency) response is dominant among all frequencies. This indicates that conducting feature distillation under such an imbalanced distribution may neglect the features of all other alternative components.
- **Parameter Space:** As the parameters of the pre-trained ViTs in the fine-tuning stage are slightly changed, their pre-training knowledge remains in the parameter space. Although the pre-training datasets are not accessible in this work, the student still has the opportunity to obtain the pre-training knowledge by aligning its parameter space to the teacher's.

Inspired by these two insightful observations, we formulate our method from two new perspectives. Based on the observation in feature space, we design *dual-view feature mimicking* (DFM, shown in Figure 1(b)), whose key insight is to complement the neglected alternative features in the KD process. Specifically, DFM employs CAP as the feature projector and incorporates two feature mimicking paths. In the first path, DFM conducts feature mimicking in the original space to learn the teacher's global features. In the second path, by removing the direct component in frequency space, DFM highlights the subtle alternative responses in feature mimicking, thus avoiding the neglect of these features. As a result, the two paths are complementary to each other, jointly promoting the feature space alignment. Based on the observation in parameter space, we propose *teacher parameter perception* (TPP, shown in Figure 1(c)), whose target is to transfer the pre-training knowledge by establishing a connection between teacher's and student's parameter spaces. Thanks to the aligned feature computing paradigm by CAP, TPP could bridge the student's early stages to the teacher's later stages and form a proxy feature processing path, where their parameter spaces join hands for KD optimization. By applying feature distillation in this path, the student's parameter space tends to be gradually aligned with the teacher's, and the pre-training knowledge would be transferred from the teacher to the student. Since the distillation learning processes in feature space and parameter space are the two sides of the same coin, DFM and TPP could naturally reinforce each other in essence.

Benefited from the progressive designs, CAP, DFM, and TPP can be seamlessly integrated into a neat and effective cross architecture knowledge distillation method, called *ScaleKD*, which addresses the above three problems as a whole. Although ScaleKD has multiple feature mimicking paths, they only exist in the training stage. That is, ScaleKD does not alter the student's structure and introduces no additional cost in the inference stage. By conducting systematic experiments on several mainstream large-scale vision benchmarks, we validate the effectiveness and generalization ability of our method.

## 2 Method

Given a pre-trained ViT teacher having $m$ stages and a target student (CNN or MLP or ViT) having $n$ stages, let $F^{s_i}$ and $F^{t_j}$ denote features from *i-th* stage of the student and *j-th* of the teacher, respectively. In what follows, we formulate all three components of ScaleKD in the form of performing feature distillation, for better clarifying their tightly coupled relationships.

### 2.1 Three Core Components in ScaleKD

**Cross Attention Projector.** As shown in Figure 1(a), CAP adopts the structure of a standard transformer decoder block, consisting of a transformer decoder layer and an MLP layer, but incorporates three critical modifications. For brevity, taking CNN as an example, our modifications include: i) patchifying regular grids of pixels in CNN; ii) adding positional embeddings; iii) setting queries in the transformer decoder block as trainable variables that share the same resolution with the teacher's features. The first two modifications intend to narrow the discrepancy between

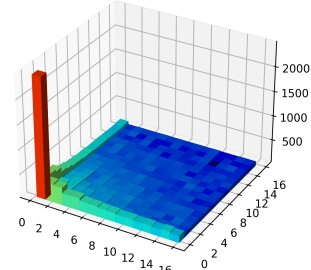

Figure 2: Feature distribution of BEiT-L/14 [41] in the frequency domain, where the direct component response is dominant. Details on drawing this figure are shown in Figure 5.

different semantic units of the pre-trained ViT teacher and the CNN student, while the last modification endows the employed transformer decoder block with great flexibility to align feature semantics and spatial resolution. For MLP and ViT students, we adopt the same CAP structure as to CNN students for simple implementation but they can adapt CAP with fewer modifications when necessary. Based on these modifications, the cross-attention operation further models global dependencies on the projected student features. With CAP, the feature distillation loss is defined as:

$$\mathcal{L}_{CAP} = \alpha L(F^t, f_p(F^s; q)) = \alpha ||F^t - f_p(F^s; q)||_2^2, \tag{1}$$

where $f_p$, $q$, $\alpha (\geq 0)$, and $L(\cdot)$ denote the CAP, the trainable queries, the loss weight, and the $L_2$-normed distance, respectively.

**Dual-view Feature Mimicking.** As shown in Figure 1(b), building upon CAP, DFM contains two feature mimicking paths. As we stated in Section 1, the first path aims to learn the teacher's global features and the second path aims to excite and mimic the alternative features (neglected by existing KD methods). Specifically, in the first path, DFM conducts feature mimicking in the teacher's original feature space, which is formulated as: $\mathcal{L}_{ori} = \alpha L(F^t, f_{p_1}(F^s, q_1))$, where $f_{p_1}$ and $q_1$ denote the CAP and its trainable queries in the first path, respectively. In the second path, the dominant direct component should be removed. To achieve this goal, we first employ discrete cosine transform (DCT), which maps the features from the spatial domain to the frequency domain: $DCT : \mathcal{X} \to \mathcal{Z}$. We then define an operator $\phi$ that removes direct component response from the features:

$$\phi(x) = DCT^{-1}(\sigma(DCT(x))) \qquad s.t. \ \sigma(z) = \begin{cases} 0, & z = 0 \\ z, & z \neq 0 \end{cases} . \tag{2}$$

Next, feature mimicking in the second path is formulated as: $\mathcal{L}_{alt} = \alpha L(\phi(F^t), \phi(f_{p_2}(F^s; q_2)))$, where $f_{p_2}$ and $q_2$ denote the CAP and its trainable queries in the second path, respectively. Now, the feature distillation loss of DFM is formulated as:

$$\mathcal{L}_{DFM} = \beta \mathcal{L}_{ori} + (1 - \beta)\mathcal{L}_{alt}, \tag{3}$$

where $\beta \in [0, 1]$ denotes the balancing weight.

**Teacher Parameter Perception.** As we stated in Section 1, TPP establishes a proxy feature processing path by connecting the student's early stages to the teacher's later stages through a CAP. In our implementation, the proxy path consists of the student's first $n-1$ stages and the teacher's last stage, as illustrated in Figure 1(c). By feature mimicking in this proxy path, the parameters of the

student part are gradually aligned with the parameters of the teacher part, thus enabling the transfer of the teacher's pre-training knowledge. Let $F^{st} = g_{t_m}(f_p^{st}(F^{s_{n-1}}; q))$ be the output features of the proxy path, where $g_{t_m}$ and $f_p^{st}$ denote the teacher's last stage and the CAP in this path, respectively. The feature mimicking in the proxy path is formulated as: $\mathcal{L}^{st} = \alpha L(F^t, F^{st})$. We further introduce $F^{st}$ as input-dependent queries for the CAP in the original path. This feature mimicking design aims to enhance the capability of CAP as such queries contain more teacher-related information, and its corresponding loss is formulated as $\mathcal{L}^s = \alpha L(F^t, f_p^s(F^{s_n}; F^{st}))$. With a simple principle of equal treatment to the two feature mimicking paths, the feature distillation loss of TPP is defined as:

$$\mathcal{L}^{TPP} = \mathcal{L}^s + \mathcal{L}^{st}. \tag{4}$$

## 2.2 Overall Formulation

From a general perspective, the progressive designs of our above three components are naturally coupled. As CAP serves as the basic component in DFM and TPP, we further introduce how to apply DFM in TPP and get a neat formulation of our method, ScaleKD. Specifically, if treating DFM as an improved version of traditional feature mimicking, it can substitute the original feature mimicking in each path of TPP. In this way, we formulate the overall design of ScaleKD, whose loss is defined as:

$$\mathcal{L}_{ScaleKD} = \mathcal{L}_{task} + \underbrace{\beta\mathcal{L}_{ori}^s + (1-\beta)\mathcal{L}_{alt}^s}_{\text{DFM for TPP Student Path}} + \underbrace{\beta\mathcal{L}_{ori}^{st} + (1-\beta)\mathcal{L}_{alt}^{st}}_{\text{DFM for TPP Teacher Path}} + \mathcal{L}_{kd}, \tag{5}$$

where $\beta \in [0, 1]$ is the balancing weight, $\mathcal{L}_{task}$ is the cross-entropy loss, and $\mathcal{L}_{kd}$ is the vanilla logits-based KD loss [14] widely used in previous KD research. As the features are standardized, we set $\alpha = 1$ for loss terms in DFM as the default. Hence, our method has only one hyper-parameter $\beta$.

## 3 Main Experiments

We perform comprehensive experiments to systematically validate the efficacy of our method and answer the question in our motivation. Specifically, our experimental verification contains six parts: i) validating the effectiveness of our method under basic settings; ii) conducting main experiments on ImageNet-1K [45] (IN-1K) dataset with various student backbones and showing the promising performance gains of our method against individually trained counterparts; iii) verifying whether our method could transfer the scalable properties of the teacher to the target student; iv) conducting transfer learning on downstream tasks with MS-COCO [46] and ADE20K [47] datasets to examine whether the performance gains from our method could be well preserved; v) comparing our method with recent top KD methods; vi) showing the potential impact of our method on model engineering.

*Unless otherwise stated, in experiments, the student backbones are trained on IN-1K from scratch, without the pre-training on other upstream datasets. Experimental details are in Appendix A and B.*

### 3.1 Pilot Experiments under Basic Settings

As we mentioned in Section 1, ScaleKD is tailored for: i) transferring the pre-trained ViT teacher's knowledge to the student having different model architectures; ii) making the student inherit the teacher's scalability. Therefore, we first perform the following two pilot experiments.

**Cross Architecture Knowledge Distillation.** To illustrate the difficulty of cross architecture feature distillation and validate the efficacy of ScaleKD under this setting, we compare ScaleKD with traditional feature distillation (FD) [15] on two different cross architecture teacher-student network pairs. From the results shown in Table 1, we can observe: i) due to architecture gaps between the teacher and the student, traditional FD shows limited performance gains; ii) comparatively, our ScaleKD achieves significantly better performance, bringing 2.75%|3.22% absolute top-1 gain for ResNet-50|Mixer-S. With the above experiments, we preliminarily verify that ScaleKD could effectively handle cross architecture feature distillation, which is difficult for traditional FD.

**Large Pre-trained ViTs as Teachers.** With ResNet-50 as the student, we examine the rationality of selecting large pre-trained ViTs as teachers in ScaleKD. Specifically, we gradually scale up the teacher's model capability (first from Swin-S to Swin-B, and then to Swin-L) and perform experiments

Table 1: Pilot experiments on cross architecture distillation with ScaleKD and FD. $s_i$ denotes the distillation is conducted on stage-i. To clearly show the performance gain, experiments in this table are conducted without $L_{kd}$.

| Teacher | Student | Method | Top-1(%) | ΔTop-1(%) |
|---|---|---|---|---|
| Swin-S (83.02) | ResNet-50 | Baseline | 76.55 | - |
| | | FD ($s_4$) | 77.43 | +0.88 |
| | | FD ($s_3,s_4$) | 77.74 | +1.19 |
| | | ScaleKD | 79.30 | +2.75 |
| | Mixer-S | Baseline | 74.02 | - |
| | | FD ($s_4$) | 74.88 | +0.86 |
| | | FD ($s_3,s_4$) | 75.32 | +1.30 |
| | | ScaleKD | 77.24 | +3.22 |

Table 2: Pilot experiments on scaling up the teacher size. The advanced training strategy uses more sophisticated data augmentation and optimizer, and longer training epochs, as shown in Table 10.

| Teacher | Student | Ratio of T/S Params | Top-1(%) | ΔTop-1(%) |
|---|---|---|---|---|
| *ScaleKD with Traditional Training Strategy* | | | | |
| *Baseline* | ResNet-50 | - | 76.55 | - |
| Swin-S (83.02) | | 1.94× | 79.62 | +3.07 |
| Swin-B (85.16) | | 3.43× | 79.80 | +3.25 |
| Swin-L (86.24) | | 7.68× | 80.10 | +3.55 |
| *ScaleKD with Advanced Training Strategy* | | | | |
| *Baseline* | ResNet-50 | - | 78.64 | - |
| Swin-S (83.02) | | 1.94× | 81.43 | +2.79 |
| Swin-B (85.16) | | 3.43× | 81.77 | +3.13 |
| Swin-L (86.24) | | 7.68× | 82.03 | +3.39 |

Table 3: Main results of ScaleKD on **11** teacher-student network pairs. † denotes the model pre-trained on IN-22K [45] and ‡ denotes the model pre-trained by EVA [41], which has the learned knowledge of LAION-2B [48].

| Teacher | Student | Params (M) | | FLOPs (G) | | Accuracy (%) | |
|---|---|---|---|---|---|---|---|
| | | T | S | T | S | Top-1 | ΔTop-1 |
| Swin-L† (86.24) | MobileNet-V1 (72.10) | 196.53 | 4.23 | 34.04 | 0.58 | 75.15 | +3.05 |
| | ResNet-50 (78.64) | | 25.56 | | 4.12 | 82.03 | +3.39 |
| | ConvNeXt-T (82.14) | | 28.59 | | 4.46 | 84.16 | +2.02 |
| | Mixer-S/16 (74.02) | 196.53 | 18.53 | 34.04 | 3.78 | 78.63 | +4.61 |
| | Mixer-B/16 (76.44) | | 59.88 | | 12.61 | 81.96 | +5.52 |
| | ViT-S/16 (79.90) | 196.53 | 22.05 | 34.04 | 4.61 | 83.93 | +4.03 |
| | Swin-T (81.18) | | 28.29 | | 4.36 | 83.80 | +2.62 |
| | ViT-B/16 (81.80) | | 86.57 | | 17.58 | 85.53 | +3.73 |
| BEiT-L/14‡ (88.58) | ResNet-50 (78.64) | 304.14 | 25.56 | 81.06 | 4.12 | 82.34 | +3.70 |
| | Mixer-B/14 (76.62) | | 59.88 | | 16.45 | 82.89 | +6.27 |
| | ViT-B/14 (82.02) | | 86.57 | | 23.09 | 86.43 | +4.41 |

using two popular training strategies. From the results shown in Table 2, we can conclude: i) ScaleKD can help the student inherit the scalability of the teacher: it can be seen that the performance gain is consistently increased under both training strategies when scaling up the teacher's model size; ii) ScaleKD can adapt to teachers' training strategies: it can be seen that our ScaleKD always brings promising performance gains, although the baseline model under the advanced training strategy gets much more competitive performance than that under the traditional training strategy.

According to the above pilot experiments, *our ScaleKD shows basic capabilities on handling cross architecture knowledge distillation from large pre-trained ViTs to CNN and MLP students.*

## 3.2 Main Results

After verifying the effectiveness of our method under our basic settings, we move forward and perform extensive experiments on more teacher-student network pairs, in order to broadly examine the scalability of our method. Specifically, we construct **11** teacher-student network pairs by choosing **2** large teachers and **10** popular models for students, covering the current mainstream architectures across ViT, MLP, and CNN.

From the results shown in Table 3, we can observe: i) in general, our ScaleKD shows great generalization ability no matter for CNN, MLP and ViT students. Over 11 teacher-student pairs, the mean top-1 accuracy improvement reaches 3.94%, and the maximum is 6.27%; ii) considering the acceleration, with Swin-L as the teacher, ResNet-50|Mixer-S/16|ViT-S/16 trained by ScaleKD even outperforms individually trained ResNet-152|Mixer-B/16|ViT-B/16 by a margin of 0.28%|2.19%|2.13%, achieving over 2.35×|3.23×|3.83× compression in terms of model size; iii) the top-1 performance gain

Table 4: Experiments on exploring scalable properties from the teacher's pre-training data. We use the best reported models with different pre-training methods as our baselines to examine whether our student model has learned the teacher's pre-training knowledge. We use Swin-L as the teacher for the first two experiments and BEiT-L/14 as the teacher for the rest two experiments. $\Rightarrow$ denotes transfer learning and * denotes the model is trained and tested with $384 \times 384$ sample resolution.

| Model | Method | Training Dataset | Dataset Samples × Epochs (M) | Viewed Samples (M) | Top-1(%) |
|---|---|---|---|---|---|
| *Supervised pre-training* | | | | | |
| ViT-B/16 | Pre-training [4] | IN-22K $\Rightarrow$ IN-1K | $13.7 \times 90 + 1.28 \times 32$ | 1274 | 83.97 |
| | | JFT-300M $\Rightarrow$ IN-1K | $300 \times 7 + 1.28 \times 32$ | 2141 | 84.15 |
| | ScaleKD | IN-1K | $1.28 \times 300$ | 384 | 85.53 |
| *Self-supervised pre-training* | | | | | |
| ViT-B/16 | BEiT [40] | IN-22K $\Rightarrow$ IN-1K | $13.7 \times 150 + 1.28 \times 100$ | 2183 | 83.70 |
| | iBOT [11] | IN-22K $\Rightarrow$ IN-1K | $13.7 \times 320 + 1.28 \times 100$ | 4512 | 84.40 |
| | ScaleKD | IN-1K | $1.28 \times 300$ | 384 | 85.64 |
| *Cross-modal pre-training* | | | | | |
| ViT-B/16 | CLIP [13] | LAION-2B $\Rightarrow$ IN-1K | $2320 \times 32 + 1.28 \times 50$ | 74304 | 85.47 |
| | | LAION-2B $\Rightarrow$ IN-12K $\Rightarrow$ IN-1K | $2320 \times 32 + 12.1 \times 60 + 1.28 \times 50$ | 75030 | 86.17 |
| ViT-B/14 | ScaleKD | IN-1K | $1.28 \times 300$ | 384 | 86.43 |
| *EVA hybrid pre-training (MIM distillation from the cross-modal pre-trained teacher)* | | | | | |
| EVA02-S/14* | EVA-02 [49] | IN-22K $\Rightarrow$ IN-1K | $13.7 \times 240 + 1.28 \times 50$ | 3352 | 85.80 |
| | ScaleKD | IN-1K | $1.28 \times 300$ | 384 | 86.22 |

Table 5: Transfer learning results (%) on MS-COCO.

| Framework | Backbone | Pre-training | Classification (IN-1K) Top-1 | Object Detection (COCO) | | | | Instance Segmentation (COCO) | | | |
|---|---|---|---|---|---|---|---|---|---|---|---|
| | | | | $AP$ | $AP_S$ | $AP_M$ | $AP_L$ | $AP$ | $AP_S$ | $AP_M$ | $AP_L$ |
| Mask R-CNN | ResNet-50 | *Baseline* | 78.64 | 40.2 | 23.0 | 44.3 | 52.5 | 37.1 | 18.0 | 40.1 | 54.9 |
| | | Ours | 82.03 (+3.39) | 42.3 | 25.5 | 46.5 | 54.6 | 39.1 | 19.3 | 42.5 | 57.1 |
| | Swin-T | *Baseline* | 81.18 | 42.7 | 26.5 | 45.9 | 56.6 | 39.3 | 20.5 | 41.8 | 57.8 |
| | | Ours | 83.80 (+2.62) | 44.4 | 28.7 | 47.9 | 58.6 | 40.8 | 21.8 | 43.7 | 59.8 |

could be increased further, when choosing stronger teachers. For instance, ScaleKD brings 0.32% additional gain to ResNet-50 when changing the pre-trained ViT teacher from Swin-L to BEiT-L/14.

## 3.3 The Scalable Properties from Teacher's Pre-training Data

As we introduced in Section 1, the ViT's performance scalability is related to two factors: model scale and pre-training data scale. The experiments in Section 3.1 and 3.2 have validated that our method could help the student inherit the positive performance effect from increasing the teacher's model scale. In this subsection, we focus on exploring the second factor: *whether or not our method could help the student learn the teacher's pre-training knowledge from its massive pre-training datasets, mitigating the knowledge density gap.* To examine this, we alter our baselines to models with pre-training and propose an evaluation principle: given that only IN-1K is visible, if ScaleKD can help the student model achieve similar performance as models with upstream pre-training, the answer to the above question is *Yes*. With this principle, we design a series of experiments based on the selected teachers in Table 4: Swin-L having the pre-training knowledge of IN-22K and BEiT-L/14 having the pre-training knowledge of LAION-2B. We compare the performance of the student models trained by ScaleKD and the corresponding counterparts trained by prevailing pre-training methods.

From Table 4, we can observe that ScaleKD performs better than various pre-training methods across all four kinds. Note that the superior performance of ScaleKD is achieved conditioned on not viewing any pre-training data. In other words, ScaleKD merely views $5.58\times$, $11.75\times$, $195.39\times$, and $8.73\times$ less samples than the counterpart methods based on supervised pre-training, self-supervised pre-training, cross-modal pre-training, and hybrid pre-training. Therefore, we can summarize two promising conclusions: i) our method could help the student learn the teacher's pre-training knowledge from massive datasets and mitigate the knowledge density gap; ii) if a well pre-trained large ViT is available, our method can be a more efficient alternative to the time-intensive pre-training.

Table 7: Performance comparison with recent top-performing KD methods. Following the settings of them, the students are trained under the advanced training strategy. Best results are **bolded**.

| Model | Method | Teacher | # Epochs | Top-1 (%) |
|---|---|---|---|---|
| Swin-T | From Scratch | - | 300 | 81.18 |
| | DIST [50] | **Swin-L (86.30)** | 300 | 82.30 |
| | DiffKD [51] | **Swin-L (86.30)** | 300 | 82.50 |
| | ScaleKD | Swin-L (86.24) | 300 | **83.80** |
| ResNet-50 | From Scratch | - | 300 | 78.60 |
| | DIST[50] | Swin-L (86.30) | 450 | 80.20 |
| | DiffKD [51] | Swin-L (86.30) | 450 | 80.50 |
| | OFA [36] | **ViT-B (86.53)** | 300 | 81.33 |
| | ScaleKD | Swin-L (86.24) | 300 | **82.03** |
| | FunMatch [52] | BiT-Res152x2 (N/A) | 1200 | 81.54 |
| | FunMatch [52] | BiT-Res152x2 (N/A) | 9600 | 82.31 |
| | ScaleKD | Swin-L (86.24) | 600 | **82.55** |

Table 8: Performance comparison with model engineering methods. More comparisons are shown in Table 14 in the Appendix.

| Model | Params (M) | FLOPs (G) | Top-1 (%) |
|---|---|---|---|
| *CNN-based Architecture* | | | |
| ResNet-50 [2] | 22.56 | 4.12 | 78.64 |
| ResNet-50 + ScaleKD | 22.56 | 4.12 | 82.55 |
| ConvNext-T [3] | 28.59 | 4.46 | 82.14 |
| RepViT-2.3M [53] | 22.90 | - | 82.50 |
| *MLP-based Architecture* | | | |
| Mixer-B/16 [6] | 59.88 | 12.61 | 76.44 |
| Mixer-B/16 + ScaleKD | 59.88 | 12.61 | 81.96 |
| gMLP-B [8] | 73.00 | 15.80 | 81.60 |
| ResMLP-B24 [7] | 115.7 | 23.00 | 81.00 |
| *ViT-based Architecture* | | | |
| ViT-S/16 [4] | 22.05 | 4.61 | 79.90 |
| ViT-S/16 + ScaleKD | 22.05 | 4.61 | 83.93 |
| Swin-T [5] | 73.00 | 15.80 | 81.18 |
| Swin-B [5] | 87.77 | 15.14 | 83.50 |

## 3.4 Transferring to Downstream Tasks

To further examine whether the performance gains from our method could be well preserved in transfer learning, we conduct comparative experiments on MS-COCO for object detection and instance segmentation, and on ADE20K for semantic segmentation.

Table 6: Transfer learning results (%) on ADE20K.

| Framework | Backbone | Pre-training | IN-1K (Top-1) | ADE20K (mIOU) |
|---|---|---|---|---|
| UperNet | ResNet-50 | *Baseline* | 78.64 | 42.37 |
| | | Ours | 82.03 (+3.39) | 44.50 (+2.13) |
| | Swin-T | *Baseline* | 81.18 | 44.41 |
| | | Ours | 83.80 (+2.62) | 46.33 (+1.92) |
| | ViT-B/16 | *Baseline* | 81.80 | 46.75 |
| | | Ours | 85.53 (+3.73) | 50.84 (+4.09) |

The results on MS-COCO and ADE20K are shown in Table 5 and Table 6, respectively, from which we can observe: i) overall, our pre-trained models outperform their baselines by significant margins across three downstream tasks and different architectures; ii) for semantic segmentation on ADE20K, ViT-B/16 achieves the highest 4.09% absolute performance gain across three backbones, even higher than its gain on IN-1K; iii) for object detection and instance segmentation on MS-COCO, ResNet-50|Swin-T pre-trained by ScaleKD outperforms its baseline by an *AP* margin of 2.1%|1.7% and 2.0%|1.5%, respectively. The above observations illustrate that the performance gains from ScaleKD could be well transferred to various and challenging downstream tasks.

## 3.5 Comparison with Recent Top-Performing KD Methods

As we stated in Section 1 and 2, ScaleKD is a unified design incorporating three novel focuses to align computing paradigm differences, model scale differences, and knowledge density differences, which are clearly different from existing KD methods. In order to validate the superiority of our method, we compare ScaleKD with recent top-performing KD methods.

From the results shown in Table 7, we can see: i) compared to DIST, DiffKD and OFA, although our teacher is not the best and the number of training epochs is the smallest, our ScaleKD still outperforms the best of these methods by clear margins (0.70%|1.30% on ResNet-50|Swin-T); ii) compared to FunMatch, our method even shows superior performance, outperforming FunMatch by a margin of 0.24% but only using less than 10% training epochs. As a result, in the context of transferring the scalability of the pre-trained ViT to various student models, our systematic design and its focuses show obvious superiority to previous works, paving a new path for future KD research.

## 3.6 Potential Impact on Model Engineering

In Section 3.2, we have noticed ScaleKD brings significant performance gains to target students, especially for the plain design in each model category, such as ResNet, MLP-Mixer, and ViT. In parallel, model engineering is a common solution to improve the model performance. Considering these two facts, we conjure that since our method could bring competitive performance gain compared to model engineering, larger flexibility would be provided when choosing models in practice.

Table 9: Ablation studies. Experiments in (b)-(d) are performed on Swin-S→ResNet-50. As DFM and TPP are designed based on CAP, CAP is added by default when choosing DFM and TPP in (a). Because of this, we treat CAP as another baseline method, when analyzing DFM and TPP in (c)-(d).

(a) Ablation on the overall design

| Teacher | Student | Ablation Design | | | | Top-1 |
| | | CAP | DFM | TPP | KD | |
|---|---|---|---|---|---|---|
| Swin-S | ResNet-50 | | | | | 76.55 |
| | | ✓ | | | | 77.87 |
| | | ✓ | ✓ | | | 78.51 |
| | | ✓ | | ✓ | | 78.62 |
| | | ✓ | ✓ | ✓ | | 79.30 |
| | | ✓ | ✓ | ✓ | ✓ | 79.62 |
| | Mixer-S | | | | | 74.02 |
| | | ✓ | | | | 75.03 |
| | | ✓ | ✓ | | | 76.42 |
| | | ✓ | | ✓ | | 76.28 |
| | | ✓ | ✓ | ✓ | | 77.24 |
| | | ✓ | ✓ | ✓ | ✓ | 77.59 |

(b) Ablation on CAP

| Projector | Top-1 | Δ Top-1 |
|---|---|---|
| *Baseline* | 76.55 | - |
| Linear | 77.43 | +0.88 |
| Conv | 77.52 | +0.97 |
| CAP | 77.87 | +1.32 |

(c) Ablation on DFM

| Feature Mimicking | Top-1 | ΔTop-1 |
|---|---|---|
| *Baseline* | 76.55 | - |
| CAP | 77.87 | +1.32 |
| Dual-Path CAP | 78.12 | +1.57 |
| DFM | 78.51 | +1.96 |

(d) Ablation on TPP

| Method | TPP Design | | Accuracy (%) | |
| | Proxy Path | Adaptive Queries | Top-1 | ΔTop-1 |
|---|---|---|---|---|
| *Baseline* | - | - | 76.55 | - |
| CAP | - | - | 77.87 | +1.32 |
| TPP | ✓ | - | 78.50 | +1.95 |
| TPP | ✓ | ✓ | 78.62 | +2.07 |

To study it, we apply ScaleKD to 3 standard designs of CNN, MLP and ViT, and compare the performance with recent advanced designs. From the results shown in Table 8, we observe that our method could help these models reach better performance than advanced models. More interestingly, in Table 3, we can clearly see that the performance gap between plain designs (ResNet-50|ViT-S/16) and advanced designs (ConvNeXt-T|Swin-T) no longer exists after applying ScaleKD. These phenomena indicate ScaleKD could have a potential impact on the model selection in real applications.

# 4 Ablation Study

## 4.1 Tightly Coupled Design Properties of Three Core Components

Recall that our ScaleKD consists of three core components, CAP, DFM and TPP, which are progressively designed in a tightly coupled manner. In Table 9a, we perform an ablation study to testify their complementarity via comparing different component combinations. We can notice: i) when gradually applying more of three component designs, the performance of ResNet-50 and Mixer-S shows similar increasing trends, showing that each component of ScaleKD is not designed for specific student architecture; ii) although CAP brings the two students promising performance gains, DFM and TPP further brings ResNet-50|Mixer-S extra performance gains, 0.64%|0.75% and 1.25%|1.39% respectively, verifying that DFM and TPP are complementary to CAP; iii) when using DFM and TPP together, both ResNet-50 and Mixer-S obtain additional performance boosts, which indicates that DFM and TPP are also complementary with each other.

## 4.2 Role of Each of Three Core Components

**CAP vs. Popular Feature Projectors.** We first compare CAP with two popular feature projectors, denoted as *Linear* and *Conv*, to verify the superiority of CAP. The former projector consists of a linear layer and the latter projector consists of two 3×3 convolutional layers. From the results shown in Table 9b, we can notice that CAP outperforms the other two projectors clearly, which validates the key role of CAP: aligning computing paradigm differences towards better KD performance.

**Importance of Alternative Feature Mimicking in DFM.** The key insight of DFM is to complement the neglected alternative features in the feature mimicking process. In Table 9c, we compare DFM with CAP and dual-path CAP to illustrate that the alternative feature mimicking is essential. We find that although the dual-path feature mimicking brings 0.25% extra performance gain to CAP, removing

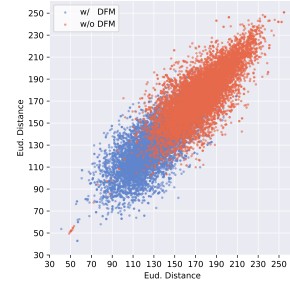

Figure 3: Feature distances of alternative components in the spatial domain. Details on the figure drawing are in Figure 6.

the direct component in the second path can further bring 0.39% improvement. This verifies the rationality of the design. To better understand why the performance gain is from the learning of teacher's alternative features, we make comparisons between the methods with DFM and without DFM. Specifically, we measure the distances between the student's features and the teacher's features of alternative components in the spatial domain. In Figure 3, we can clearly see that DFM can effectively reduce alternative feature distance between the teacher and the student.

**Role of Proxy Path in TPP.** Note that for transferring the teacher's pre-training knowledge in the parameter space to the student, TPP establishes a proxy path that connects the student's former stages to the teacher's later stages. In Table 9d, we study the design of TPP and verify whether the proxy path and its adaptive queries are effective. The results show that the feature mimicking in the proxy path can provide the student with performance improvement and providing input-dependent queries can further enhance the effectiveness of TPP, which indicates that these designs in TPP are essential for learning the knowledge in the teacher's parameter space.

*More ablation studies on the hyper-parameter $\beta$ and the others are provided in Appendix D.*

## 5    Related Work

**Knowledge Distillation.** Traditional KD methods [14–35] generally focus on CNN-based teacher-student network pairs with small model scale gaps. Some recent works [54, 55, 50] further study how to conduct knowledge distillation with larger teachers. As vision transformers suffer from low convergence speeds, some recent works [56–58] explore leveraging CNNs to accelerate the training of vision transformers. Meanwhile, [36–38, 59] discuss how to bridge the architecture gap when the teacher and the student are in different model categories.

**Frequency-based Knowledge Distillation.** As traditional feature distillation only focuses on pixel-to-pixel differences, FAM [60] defines knowlwedge distillation in terms of frequency-based attention maps. FreeKD [61] explores how to eliminate unfavorable information in the frequency domain for enhancing the distillation performance on dense prediction tasks. Different from our ScaleKD, they consider feature distillation on CNN-based network pairs and have different formulations.

**Teacher Parameter Reuse.** Some previous KD methods also leverage the teacher's parameter for reusing a better classifier [32] or initializing the student's neck and head [62–64] or dismissing the shortcuts in residual architectures [65]. Unlike our ScaleKD, the motivation of these works focuses on parameter reuse or equivalent substitution, rather than aligning two parameter spaces for transferring the teacher's pre-training knowledge to the target student without the pre-training process.

## 6    Conclusion

In this paper, we present ScaleKD, a new cross architecture KD approach for transferring the scalable properties of pre-trained large ViTs to various CNNs, MLPs and heterogeneous ViTs. Our method consists of three tightly coupled components that rely on principled designs to align computing paradigm differences, model scale differences, and knowledge density differences between the teacher and the student. By conducting systematic experiments on several mainstream large-scale vision benchmarks, we broadly validate the effectiveness and generalization ability of our method. Benefiting from its novel motivation and design insights, ScaleKD is the first work which successfully verified that KD can be a more efficient alternative to the time-intensive pre-training, to the best of our knowledge. This extends the application scope of KD from model compression to training acceleration. We hope our work would inspire feature KD research in this new direction.

**Limitations.** Restricted by our computational resources, we do not conduct experiments on very large teachers, such as ViT-22B [66], or on large students, such as ViT-L [4]. Furthermore, with the increasing model scale of teachers, the training cost of ScaleKD increases, which is a common limitation to KD research. According to the analysis in Appendix D, the extra training cost of ScaleKD is acceptable to a large extent. Actually, thanks to its promising performance, ScaleKD shows the great potential to replace the time-intensive pre-training of students on large-scale datasets.

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

# Appendix for "ScaleKD: Strong Vision Transformers Could Be Excellent Teachers"

**Jiawei Fan**[*]
Intel Labs China
jiawei.fan@intel.com

**Chao Li**[*]
Intel Labs China
chao3.li@intel.com

**Xiaolong Liu**[*]
iMotion Automotive Technology
xiaolong.liu@imotion.ai

**Anbang Yao**[*†]
Intel Labs China
anbang.yao@intel.com

## A  Datasets

**ImageNet-1K [45]** is a well-known large-scale classification dataset, comprising over 1.2 million training images and 50,000 validation images with 1,000 object categories.

**MS-COCO [46]** is a large-scale dataset for object detection and instance segmentation, which contains 118,000 training images and 5,000 validation images with 80 object categories.

**ADE20K [47]** is a challenging semantic segmentation dataset, containing 20,210 training samples, 2,000 validation samples, and 3,352 testing samples with 150 categories.

## B  Experimental Setups

### B.1  Experimental Setups on IN-1K

**Training Strategy.** We conduct our experiments with two popular training strategies: traditional training strategy and advanced training strategy. The traditional training strategy is commonly used in previous KD approaches (shown in Table 10a) and the advanced training strategy is adopted in training recently proposed CNNs, MLPs, and ViTs (shown in Table 10b).

**Compute Infrastructure.** The experiments using the traditional training strategy are conducted on $8 \times$ NVIDIA Tesla-V100 GPUs, while the experiments using the advanced training strategy are conducted on $32 \times$ NVIDIA Tesla-V100 GPUs.

**Implementation Codebase.** We implement our method based on MMClassification [67].

**Hyper-Parameter Settings.** The overall loss for our ScaleKD is defined as Equation 5. Thanks to the simplicity of this formulation, we have only one hyper-parameter $\beta$ in our ScaleKD. From the ablation study in the Appendix D, we find that the best choice is $\beta = 0.6$ and we use it as the default setting throughout all experiments.

**Selection of Teacher-Student Network Pairs.** Overall, we construct 11 teacher-student network pairs, which consist of 2 pre-trained large ViTs, and 10 students covering mainstream architectures of ViT, MLP, and CNN. Specifically, for the teacher, we choose two different types of well pre-trained ViTs: supervised pre-trained Swin-L [5] with the hierarchical architecture and hybrid pre-trained BEiT-L/14 [40] with the typical ViT architecture. Moreover, compared to Swin-L, BEiT-L/14 is much larger in terms of model size and stronger in terms of model performance. For the student

---

[*] Core authors contributed to method formulation, experimental design and analysis.

[†] Corresponding author.

Table 10: Detailed settings of traditional training strategy and advanced training strategy on IN-1K.

(a) Traditional training strategy

| Configuration | CNN |
|---|---|
| Batch Size | 256 |
| Learning Rate | 0.1 |
| Learning Rate Schedule | Stepwise Decay/Cosine Decay |
| Optimizer | SGD |
| Optimizer Hyper-Parameters | momentum= 0.9 |
| Weight Decay | 1e-4 |
| Training Epochs | 100 |
| Warmup Epochs | ✗ |
| Drop Path | ✗ |
| Label Smoothing | ✗ |
| Random Flip | 0.5 |
| Random Resize Crop | (0.08,1) |
| Random Augmentation | ✗ |
| Random Erasing | ✗ |

(b) Advanced training strategy

| Configuration | CNN / MLP / ViT |
|---|---|
| Batch Size | 2048 / 1536 / 1024 |
| Learning Rate | 5e-3 / 7e-4 / 1e-3 |
| Learning Rate Schedule | Cosine Decay |
| Optimizer | Lamb / AdamW / AdamW |
| Optimizer Hyper-Parameters | $\beta_1, \beta_2, \epsilon = 0.9, 0.009, 1e-8$ |
| Weight Decay | 0.02 / 0.07 /0.05 |
| Training Epochs | 300 |
| Warmup Epochs | 5 / 20 / 20 |
| Drop Path | 0.05 / 0.1 / 0.1 |
| Label Smoothing | 0.1 |
| Random Flip | 0.5 |
| Random Resize Crop | (0.08,1) |
| Random Augmentation | (7,0.5) / (9,0.5) / (9,0.5) |
| Random Erasing | 0.25 |

Table 11: Detailed settings of transfer learning strategies on MS-COCO and ADE20K.

(a) MS-COCO

| Configuration | ResNet-50 / Swin-S |
|---|---|
| Weight Initialization | Pre-trained Checkpoint |
| Batch Size | 16 |
| Learning Rate | 1e-4 |
| Learning Rate Decay | Stage (0.7) |
| Learning Rate Schedule | Cosine Decay |
| Optimizer | AdamW |
| Optimizer Hyper-Parameters | $\beta_1, \beta_2, \epsilon = 0.9, 0.009, 1e-8$ |
| Weight Decay | 0.05 |
| Training Epochs | 8 |
| Crop Size | (1333, 800) |
| Drop Path | 0.0 / 0.2 |

(b) ADE20K

| Configuration | ResNet-50 / Swin-S / ViT-B |
|---|---|
| Weight Initialization | Pre-trained Checkpoint |
| Batch Size | 16 |
| Learning Rate Decay | 1e-4 / 1e-4 / 2e-4 |
| LR decay | Stage (0.9) / Stage (0.9) / Layer (0.6) |
| Learning Rate Schedule | Cosine Decay |
| Optimizer | AdamW |
| Optimizer Hyper-Parameters | $\beta_1, \beta_2, \epsilon = 0.9, 0.009, 1e-8$ |
| Weight Decay | 0.05 |
| Training Iterations | 160000 |
| Crop Size | (512, 512) |
| Drop Path | 0.0 / 0.3 / 0.2 |

architectures, we first choose the basic design in each architecture type, such as ResNet-50 [2], Mixer-S/16 [6], and ViT-S/16 [4]. Then, we also select some popular models, such as MobileNet-V1 [68], ConvNeXt-T [3], and Swin-S. Next, we expand the basic designs to larger ones, like Mixer-B/16, Mixer-B/14, ViT-B/16 and ViT-B/14. After separately selecting teachers and students, we finally organize them into 11 teacher-student network pairs for comprehensive experiments. Note that the performance for most individual trained baselines in Table 3 are sourced from their original papers, except for ResNet-50, MobileNet-V1, ViT-B/14, and Mixer-B/14, which we trained ourselves using the advanced training strategy due to the absence of reference results in their original papers.

**Counterpart Pre-training Methods.** In the main paper, we select state-of-the-art methods in each pre-training paradigm for comparison. For supervised pre-training, we choose the pioneering work [4]. For self-supervised pre-training, we choose BEiT [40] and iBoT [11]. For cross-modal pre-training and hybrid pre-training, we choose CLIP [13] and EVA-02 [49], respectively.

**Counterpart Knowledge Distillation Methods.** In the main paper, we make comparisons with many recent KD methods, such as DIST [50], DiffKD [51], OFA [36] and FuncMatch [52]. In this Appendix, we further compare with CNN-based methods, such as KD [14], AT [16], OFD [26], RKD [20], CRD [25], DKD [35], SRRL [30], ReviewKD [69], DistPro [33] and MGD [70].

**Counterpart Model Engineering Methods.** In the main paper, to better show the great potential of our ScaleKD, we apply it to the popular designs of each architecture type and make comparisons with various advanced counterparts. Driven by this target, we mainly select the so-called next-generation models. For ResNet-50, we select ConvNeXt-T [3] and RepViT-2.3M [53] for comparison. The former one is the typical design of the new-era CNN and the latter one is a popular model for deployment. For Mixer-B, we select gMLP-B [8] and ResMLP-B24 [7], which are optimized to

Table 12: Performance comparison (%) of ScaleKD and CLIP on ViT-B for linear probing.

| Model | Method | Pre-training Dataset | IN-1K (Training) | CIFAR-100 (Linear Probing) | | | | |
|-------|--------|---------------------|------------------|--------|--------|---------|---------|------|
| | | | | 1 shot | 5 shot | 10 shot | 25 shot | Full |
| ViT-B/16 | From-scratch | IN-1K | 81.80 | 33.86 | 60.30 | 66.77 | 72.65 | 81.76 |
| | CLIP | LAION-300M | - | - | - | 71.96 | 77.21 | 84.07 |
| | | LAION-2B, IN-1K | 85.49 | 41.10 | 69.00 | 72.34 | 78.64 | 85.51 |
| | | LAION-2B, IN-12K, IN-1K | 86.17 | 44.90 | 70.19 | 76.77 | 81.43 | 88.88 |
| | | CLIP OpenAI, IN-12K, IN-1K | 85.99 | 47.40 | 70.85 | 77.37 | **81.52** | 88.92 |
| ViT-B/14 | Ours | IN-1K | 86.43 | **48.14** | **70.91** | **77.52** | 81.50 | **89.11** |
| Teacher: BEiT-L/14 | EVA | CLIP OpenAI, IN-22K, IN-1K | 88.58 | 63.74 | 85.20 | 87.39 | 89.27 | 93.36 |

Table 13: Performance comparison on IN-1K with more CNN-based KD methods. In the experiment, we adopt the same traditional training strategy as these methods.

| Model | Teacher | Method | Top-1(%) |
|-------|---------|--------|----------|
| MobileNet-V1 | ResNet-50 (76.16) | From Scratch | 69.63 |
| | | KD [14] | 70.68 |
| | | AT [16] | 70.72 |
| | | OFD [26] | 71.25 |
| | | RKD [20] | 71.23 |
| | | CRD [25] | 71.40 |
| | | DKD [35] | 72.05 |
| | | SRRL [30] | 72.49 |
| | | ReviewKD [69] | 72.56 |
| | | DIST [50] | 73.24 |
| | | DistPro [33] | 73.26 |
| | | MGD [70] | 73.35 |
| | | DiffKD [51] | 73.62 |
| | Swin-L (86.24) | ScaleKD | **74.21** |

suppress the weaknesses of MLP-Mixer. For ViT-S, we choose Swin-S [5] and Swin-B as counterparts to validate whether our ScaleKD could outperform larger advanced designs.

## B.2 Experimental Setups on MS-COCO and ADE20K

**Training Strategy and Hyper-Parameter Settings.** For the experiments on MS-COCO, we adopt the settings shown in Table 11a, while for experiments on ADE20K, we adopt the settings shown in Table 11b.

**Compute Infrastructure.** All experiments on MS-COCO and ADE20K are conducted on 8 × NVIDIA Tesla-V100 GPUs.

**Implementation Codebase.** We conduct experiments based on MMDetection [71] and MMSegmentation [72].

**Selection of Task Frameworks and Backbones.** For different task frameworks, we choose Mask R-CNN [73] for object detection and instance segmentation, and UperNet [74] for semantic segmentation. As for backbones, we select ResNet-50, Swin-T, and ViT-B/16.

## C  More Experiments

**Linear Probing on CIFAR-100.** We conduct a set of linear probing experiments on CIFAR-100 [75], based on models in Table 4. From the results shown in Table 12, we can observe: i) models pre-trained by CLIP greatly improve the backbone's generalization ability across different datasets; ii) our ScaleKD helps the student model reach mostly better performance than CLIP-based pre-training, even without viewing pre-training data.

**Performance Comparison with More KD Methods.** In the main paper, we compare ScaleKD with mostly related cross architecture KD approaches in Table 7, as few previous works use medium-sized students, such as ResNet-50, for benchmarking. To make a more comprehensive comparison with lots of CNN-based KD methods, we conduct experiments on a traditional student network, MobileNet-V1,

Table 14: Performance comparison on IN-1K with lots of model engineering methods. We conduct ScaleKD on the simplest design of each architecture type and then make a performance comparison with various designs. Ours$^\star$, Ours$^\dagger$ and Ours$^\ddagger$ denote choosing ViT-B (training from scratch), Swin-L (with IN-22K pre-training) and BEiT-L/14 (with EVA pre-training) as the teacher, respectively.

| Model | Training Dataset | Resolution | Params (M) | FLOPs (G) | Top-1(%) |
|---|---|---|---|---|---|
| *CNN-based Architecture* | | | | | |
| ConvNext-T [3] | IN-1K | $224^2$ | 28.59 | 4.46 | 82.14 |
| ConvNext-T + Ours$^\dagger$ | IN-1K | $224^2$ | 28.59 | 4.46 | 84.16 |
| ConvNext-T [3] | IN-22K $\Rightarrow$ IN-1K | $224^2$ | 28.59 | 4.46 | 82.90 |
| ConvNext-B [3] | IN-1K | $224^2$ | 87.77 | 15.14 | 83.80 |
| UniRepLKNet-T [76] | IN-1K | $224^2$ | 31.00 | 4.90 | 83.20 |
| EfficientNet-B5 [77] | IN-1K | $456^2$ | 30.00 | 9.90 | 83.60 |
| RepViT-M2.3 [53] | IN-1K | $224^2$ | 22.90 | - | 83.70 |
| *MLP-based Architecture* | | | | | |
| Mixer-B/16 [6] | IN-1K | $224^2$ | 59.88 | 12.61 | 76.44 |
| Mixer-B/16 + Ours$^\star$ | IN-1K | $224^2$ | 59.88 | 12.61 | 81.62 |
| Mixer-B/16 + Ours$^\dagger$ | IN-1K | $224^2$ | 59.88 | 12.61 | 81.96 |
| Mixer-B/14 + Ours$^\ddagger$ | IN-1K | $224^2$ | 59.88 | 16.45 | 82.89 |
| Mixer-B/16 [6] | IN-22K $\Rightarrow$ IN-1K | $224^2$ | 59.88 | 12.61 | 80.64 |
| Mixer-L/16 [6] | IN-22K $\Rightarrow$ IN-1K | $224^2$ | 208.2 | 44.57 | 82.89 |
| ResMLP-B24 [7] | IN-1K | $224^2$ | 115.7 | 23.0 | 81.00 |
| gMLP-B [8] | IN-1K | $224^2$ | 73.00 | 15.80 | 81.60 |
| *Transformer-based Architecture* | | | | | |
| ViT-S/16 [4] | IN-1K | $224^2$ | 22.05 | 4.61 | 79.90 |
| ViT-S/16 + Ours$^\dagger$ | IN-1K | $224^2$ | 22.05 | 4.61 | 83.93 |
| ViT-S/16 [4, 78] | IN-22K $\Rightarrow$ IN-1K | $224^2$ | 22.05 | 4.61 | 80.50 |
| Swin-T [5, 78] | IN-22K $\Rightarrow$ IN-1K | $224^2$ | 28.29 | 4.36 | 81.90 |
| T2T-ViT$_t$-14 [79] | IN-1K | $224^2$ | 21.47 | 4.34 | 81.83 |
| DaViT-T [80] | IN-1K | $224^2$ | 28.36 | 4.54 | 82.24 |
| iLLaMA-S [81] | IN-1K | $224^2$ | 21.90 | - | 79.90 |
| EVA-02-S/14 [49] | IN-1K | $336^2$ | 22.13 | 15.51 | 81.12 |
| EVA-02-S/14 + Ours$^\ddagger$ | IN-1K | $336^2$ | 22.13 | 15.51 | 86.22 |
| ViT-B/16 [4] | IN-1K | $224^2$ | 86.57 | 17.58 | 81.80 |
| Swin-B [5] | IN-1K | $224^2$ | 87.77 | 15.14 | 83.50 |
| T2T-ViT$_t$-24 [79] | IN-1K | $224^2$ | 64.00 | 12.69 | 82.71 |
| DaViT-B [80] | IN-1K | $224^2$ | 87.95 | 15.51 | 84.09 |
| ViT-B/16 [4] | IN-22K $\Rightarrow$ IN-1K | $224^2$ | 86.57 | 17.58 | 83.97 |
| Swin-B [5] | IN-22K $\Rightarrow$ IN-1K | $224^2$ | 87.77 | 15.14 | 85.20 |
| iLLaMA-B [81] | IN-22K $\Rightarrow$ IN-1K | $224^2$ | 86.30 | - | 85.00 |

using the same training strategy as them. From the results shown in Table 13, we can observe that by using Swin-L as the teacher, our ScaleKD could help MobileNet-V1 reach 74.21% top-1 accuracy, outperforming previous methods which use ResNet-50 as the teacher by clear margins.

**Performance Comparison with More Model Engineering Methods.** In the main paper, we apply ScaleKD to the basic design of each architecture type and make comparisons with more recent variant architectures. As illustrated in Table 14, we choose more designs to have a more comprehensive comparison.

## D   More Ablation Studies

**Ablation Study on Training Efficiency of ScaleKD.** As TPP in our ScaleKD leverages the teacher's last stage, it will introduce additional training costs compared to traditional FD. To clearly study its training efficiency, we conduct ablative experiments in this section: i) as shown in Table 15a, we first compare the training efficiency of ScaleKD with traditional FD on three network pairs having increased teacher's model scale; ii) then, as shown in Table 15b, we conduct the experiments on each component in ScaleKD. The experimental results show that: i) using large teachers would

Table 15: Experiments on the training efficiency of ScaleKD. The student model in all experiments is ResNet-50. In (a), we compare ScaleKD with traditional FD using three teachers with different model scales. In (b), we conduct the experiments based on Swin-S→ResNet-50 teacher-student network pair to illustrate the training costs (memory and time) introduced by each component of ScaleKD. Experiments are conducted on $8 \times$ NVIDIA Tesla-V100 GPUs.

(a) Training costs comparison with FD

| Teacher | Method | Top-1 (%) | GPU Memory (G) | $T_{train}$ (d) |
|---|---|---|---|---|
| Swin-S | FD | 77.43 | 3.66 | 1.67 |
| | ScaleKD | 79.62 | 6.65 | 2.10 |
| Swin-B | FD | 77.76 | 3.66 | 1.83 |
| | ScaleKD | 79.80 | 7.26 | 2.53 |
| Swin-L | FD | 77.72 | 4.72 | 2.24 |
| | ScaleKD | 80.10 | 9.11 | 3.51 |

(b) Training costs of each component in ScaleKD

| Method | Designs | | | | Top-1 (%) | GPU Memory (G) | $T_{train}$ (d) |
| | CAP | DFM | TPP | KD | | | |
|---|---|---|---|---|---|---|---|
| FD | - | - | - | - | 77.43 | 3.66 | 1.67 |
| ScaleKD | ✓ | | | | 77.87 | 3.77 | 1.70 |
| | ✓ | ✓ | | | 78.51 | 4.02 | 1.77 |
| | ✓ | | ✓ | | 78.62 | 5.13 | 1.84 |
| | ✓ | ✓ | ✓ | | 79.30 | 6.60 | 2.08 |
| | ✓ | ✓ | ✓ | ✓ | 79.62 | 6.65 | 2.10 |

Table 16: Ablation study on pre-training and distillation.

| Model | Method | Top-1(%) |
|---|---|---|
| ViT-S/16 | Training from scratch on IN-1K | 79.90 |
| | Training from scratch on IN-1K w/ KD | 81.42 |
| | Pre-training on IN-22K | 80.05 |
| | Pre-training on IN-22K w/ KD | 82.00 |
| | Training from scratch on IN-1K w/ ScaleKD | 83.93 |

Table 17: Ablation study on the necessity of alternative components in the first path of DFM.

| Method | Top-1 (%) | Δ Top-1 (%) |
|---|---|---|
| *Baseline* | 76.55 | - |
| CAP | 77.87 | +1.32 |
| DFM (Dir + Alt) | 78.23 | +1.68 |
| DFM (All + Alt) | 78.51 | +1.96 |

induce more GPU memory occupation and longer training time; ii) comparatively, TPP is the most resource-consuming component, especially after combining it with DFM. In summary, our ScaleKD introduces additional training costs compared to traditional FD. However, if considering the significant performance gain it brings, these additional costs are acceptable.

**Ablation Study on Hyper-parameter $\beta$.** According to Equation 5 in the main paper, our method only has one hyper-parameter $\beta$, which is the balancing weight of two feature mimicking paths in DFM. We conduct the ablation study on Swin-S→ResNet-50 network pair to study the impact of different settings of $\beta$. Specifically, we select the $\beta$ uniformly from 0 to 1. $\beta = 1.0$ indicates that only the first feature mimicking path exists, while $\beta = 0$ indicates that only the second feature mimicking path exists. As shown in Figure 4, we can observe: i) in general, ScaleKD outperforms the baseline by significant margins at all settings, validating the stability of ScaleKD; ii) when $\beta = 0.6$, our ScaleKD achieves the best performance; iii) though the second feature mimicking path could be used individually, it is inferior to the first path, indicating that the direct component is essential in feature mimicking; iv) when the two paths work collaboratively, they perform better than two individual counterparts, which suggests that the two designs are complementary with each other.

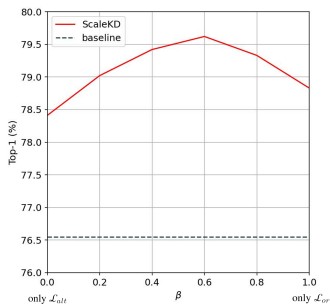

Figure 4: Ablation study on the hyper-parameter $\beta$.

**Ablation Study on Pre-training and Distillation.** In this study, we explore the originality of the distillation performance gain. We compare models trained by ScaleKD with upstream pre-trained models and upstream pre-training models with KD. From the results shown in Table 16, we can notice: i) compared to individual pre-training, applying KD under this stage can significantly boost model performance; ii) ViT-S/16 trained by ScaleKD significantly outperforms the models trained with KD on IN-22K. These two observations indicate that: i) small students are difficult to capture the pre-training knowledge with traditional FD, even with the upstream pre-training dataset; ii) our ScaleKD could effectively help the student to learn useful pre-training knowledge from the teacher without viewing the pre-training dataset.

**Ablation Study on the First Feature Mimicking Path of DFM.** We explore the necessity of alternative components in the first feature mimicking path of DFM. Specifically, we remove all alternative components in the first path of DFM and perform an experiment under the same settings as Table 9(b). For the results shown in Table 17, DFM(Dir + Alt) indicates the above new setting, while DFM(All+ Alt) is the original design. We can observe that removing all alternative components in the

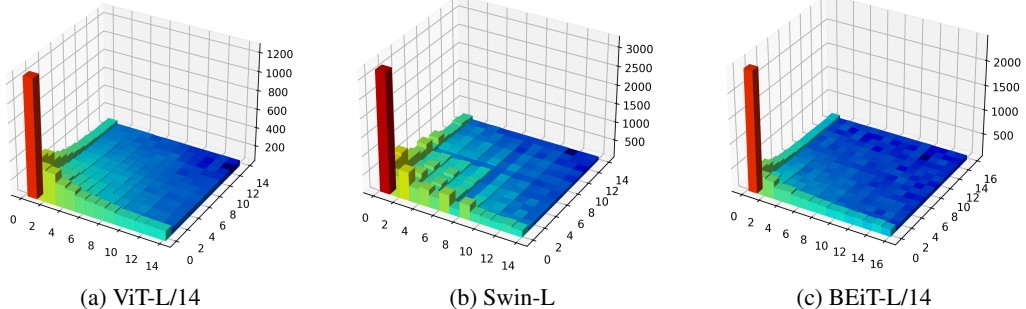

| (a) ViT-L/14 | (b) Swin-L | (c) BEiT-L/14 |

Figure 5: More illustrative feature distributions of large pre-trained ViTs in the frequency domain. We first collect the output feature maps of 1600 samples from IN-1K, then conduct DCT on each channel, and finally take the average value across these samples after converting all responses into absolute values.

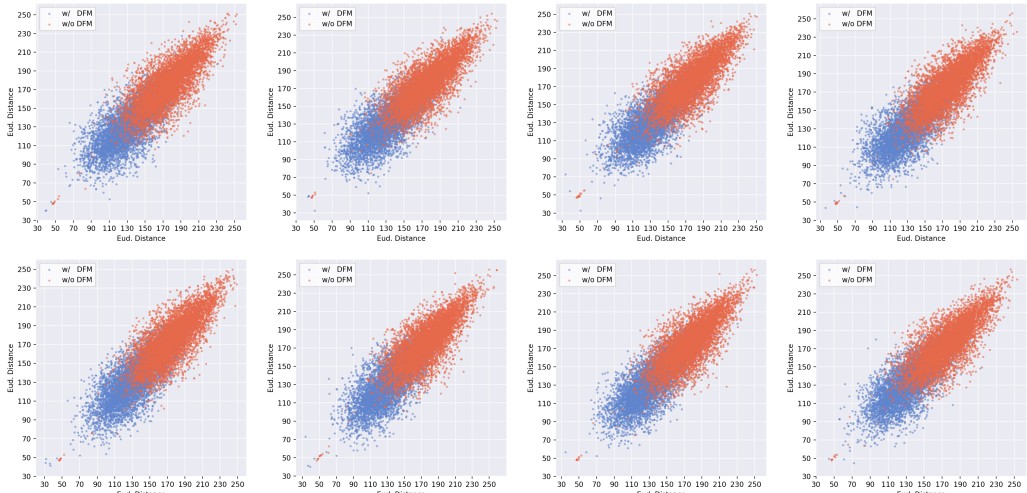

Figure 6: Feature distance distributions of alternative components for the last stage features between teacher and student on IN-1K. We obtain 64,000 feature pairs on Swin-L→ResNet-50 network pair from 64,000 samples. After calculating the distance between teacher and student, we project the high-dimension distances into a two-dimension space for illustration. Finally, we randomly select 6,400 data points for 8 times to draw the scatters. Blue points denote the distances without DFM, while orange points denote the distances with DFM.

first path will slightly decrease the effectiveness of DFM, but its performance is still obviously higher than CAP. As we discussed in Section 1, the first path of DFM aims to capture the teacher's global features, where the subtle alternative components are also indispensable parts. Directly removing alternative components in the first path will break the integrity of the original feature space (ScaleKD is not conducted in the frequency space), thus lowering the efficacy of DFM.

# E   More Visualization Results

In this section, we provide more visualization results for a better understanding of our method. In Figure 5, we provide the frequency distributions of three pre-trained large ViT models. We can observe that these pre-trained ViTs show a consistency in unbalanced frequency distributions: the direct responses are salient and significantly stronger than the alternative responses. And in Figure 6, we show more examples of feature distance distributions of alternative components, comparing scenarios with and without DFM, between the teacher and the student on IN-1K. The results validate DFM can effectively reduce the alternative feature distances between the teacher and the student.

