# OpenReview forum: "ScaleKD: Strong Vision Transformers Could Be Excellent Teachers"
_NeurIPS.cc/2024/Conference — NeurIPS 2024 poster_

### Official Review · Reviewer_mfZF · 2024-06-18

**Soundness:** 3
**Presentation:** 3
**Contribution:** 3
**Rating:** 6
**Confidence:** 4

**Summary:**

This paper introduces a novel knowledge distillation method called ScaleKD. The method aims to leverage well pre-trained vision transformer models as teacher models for a variety of student model architectures.the authors first adopt a cross attention projector to align student features with the teacher's. Then, a dual-view feature mimicking module and a teacher parameter perception module are used to achieve better knowledge transfer. Extensive experiments demonstrate the effectiveness of ScaleKD across various tasks and model types.

**Strengths:**

1. The paper is well-written and well-structured.
2. The authors provide extensive experimental results and analyses that demonstrate the effectiveness of the proposed method.

**Weaknesses:**

1. The paper's motivation could be strengthened. As highlighted in previous research [1], a stronger teacher model does not always equate to a better teacher. The necessity of adapting a ViT teacher for training a CNN student needs further justification. Including a comparison between ViT teachers and CNN teachers would provide better support.
2. The teacher parameter perception (TPP) module's cross-architecture KD paradigm is similar to techniques used in existing research [2,3]. While this does not diminish the novelty of other aspects of the paper, these related works should be properly discussed.
3. The authors critique the focus on "evaluation on small datasets with non-mainstream student models" in existing works. Some relevant papers about these issues should also be discussed, such as [4,5].

[1] Mirzadeh, Seyed Iman, et al. "Improved knowledge distillation via teacher assistant." Proceedings of the AAAI conference on artificial intelligence. Vol. 34. No. 04. 2020.

[2] Wang, Jiabao, et al. "CrossKD: Cross-head knowledge distillation for object detection." Proceedings of the IEEE/CVF Conference on Computer Vision and Pattern Recognition. 2024.

[3] Bai, Haoli, et al. "Few shot network compression via cross distillation." Proceedings of the AAAI Conference on Artificial Intelligence. Vol. 34. No. 04. 2020.

[4] Hao, Zhiwei, et al. "Vanillakd: Revisit the power of vanilla knowledge distillation from small scale to large scale." arXiv preprint arXiv:2305.15781 (2023).

[5] Stanton, Samuel, et al. "Does knowledge distillation really work?." Advances in Neural Information Processing Systems 34 (2021): 6906-6919.

**Questions:**

In the dual-view feature mimicking module, the direct component is omitted during the alternative feature mimicking process. However, the remaining non-direct features are duplicated in the normal matching process (as shown in the upper path in Figure 2b). How does the removal of the non-direct component in the upper path affect performance?

**Limitations:**

Yes

---

> ### Author Rebuttal · Authors · 2024-08-07
>
> Thank you for the recognition of our work and the constructive comments.
>
> **1. To your comments regarding strengthening the motivation of our work,**
> >**Our responses: (1)** Yes, in previous KD works like [1] using CNNs as the teachers, a stronger model is not always a better teacher. Following your suggestion, with ResNet-50 as the student, we additionally use CovNeXt-XL (the current top-performing CNN) as the teacher (adopted in recent work [4]) besides Swin-L and BEiT-L. From the results shown below, we can observe that: i) ViT teachers are superior to CNN teachers for our method, e.g., a notable 0.3%|0.62 top-1 gain is achieved when the teacher is changed from ConNeXt-XL to Swin-L|BEiT-L; ii) Given ConNeXt-XL as the teacher, our method also outperforms [4] with a clear margin. These results validate the necessity of adapting large pre-trained ViTs as teachers for training a CNN student; **(2)** Besides, we would like to emphasize the motivation of our work. Along with the evolution of network architectures and model learning paradigms, ViTs show notably improved performance when scaling up the model size and the pre-training data size. The motivation of our work is to connect cross architecture KD research with well pre-trained ViT teacher models that stand out for their remarkable scalable properties, exploring an effective way (ScaleKD) to transfer teachers' scalable properties to student models (CNNs/MLPs/heterogeneous ViTs), given that the pre-training data is invisible to student models.
> |Teacher|Student|Method|Top-1(%)|
> |--|--|:--:|:--:|
> |ConNeXt-XL(86.97)|ResNet-50(79.80)|VanillaKD|81.10
> |ConNeXt-XL(86.97)|ResNet-50(78.64)|ScaleKD|81.72
> |Swin-L(86.24)|ResNet-50(78.64)|ScaleKD|82.02
> |BEiT-L(88.58)|ResNet-50(78.64) |ScaleKD|82.34
>
> **2. To your comments regarding the discussion of our TPP module with some existing research [2,3],**
> >**Our responses: (1)** Thank you for mentioning these two works [2,3] that are related to our TPP module. Actually, we discussed some similar related works [32,59,60,61] in Line#329-333 of the manuscript, which are in the same line of research as [2,3]. Generally, TPP differs from these works in motivation and designs; **(2)** The motivation of why previous works construct the proxy path is to avoid contradictory supervision signals from the annotations and the teacher’s predictions [2], or to avoid accumulated estimation errors [3]. In contrast, the key motivation of TPP is to align the teacher’s and student’s parameter spaces, which is beneficial for pursuing the pre-training knowledge implicitly contained in the teacher's parameter space; **(3)** As the motivations are not the same, our TPP has some special modifications. On the one hand, unlike previous works that reuse a proportion of detection head [2] or a single layer [3], TPP uses the last whole stage of the teacher to have an integrated perception of teachers' parameter space. On the other hand, to have a better feature alignment, the outputs of the proxy path are provided as input-dependent queries of our CAP module.
>
> **3. To your comments regarding the discussion of our method with [4,5],**
> >**Our responses: (1)** Thank you for mentioning two related works [4,5], which aim for a better understanding of vanilla KD based on logits ([14] in our manuscript). Specifically, the early work [5] studies how to improve fidelity (measuring the predication consistency between teacher and student models) instead of the prevailing generalization ability (model accuracy), and the later work [4] explores how advanced training recipes (e.g., larger datasets, stronger data augmentations/optimizers, significantly longer training epochs (4800 epochs)) affect the performance gap of vanilla KD to its variants. Generally, [4,5] do not present any new KD methods; **(2)** We already discussed a related work FuncMatch [49] in our manuscript, which has similar statements as [4]; **(3)** In contrast, our work presents a new cross architecture KD method, showing that well pre-trained ViT models could be used as teachers and their scalable properties could be transferred to CNN/MLP/heterogeneous ViT students;**(4)** Due to the orthogonal focuses, our method may get improved performance using advanced training recipes in [4]. We leave it for future work.
>
> **4. To your question about the effect of removing the non-direct components in the first path of DFM,**
> > **Our responses: (1)** Following your comments, we remove all non-direct components in the first path of DFM and perform an experiment under the same settings as Table 9(b) in our manuscript; **(2)** For the results shown below, DFM(Dir + Alt) indicates your mentioned setting, while DFM(All+ Alt) is the original design. We can observe that removing the non-direct components in the first path will slightly decrease the effectiveness of DFM, but its performance is still obviously higher than CAP. As we discussed in Line#109-113, the first path of DFM aims to capture the teacher's global features, where the subtle non-direct components are also indispensable parts. Directly removing non-direct components in the first path will break the integrity of the original feature space (ScaleKD is not conducted in the frequency space), thus lowering the efficacy of DFM.
> |Method|Top-1(%)|$\Delta$Top-1(%)|
> |--|:--:|:--:|
> Baseline|76.55|-|
> CAP|77.87|+1.32
> DFM(Dir+Alt)|78.23|+1.68
> DFM(All+Alt)|78.51|+1.96
>
> >[1] S.I., Mirzadeh, et al. "Improved...teacher assistant", AAAI 2020.
> [2] J Wang, et al. "CrossKD...for object detection", CVPR-W 2024.
> [3] H Bai, et al. "Few shot...cross distillation", AAAI 2020.
> [4] Z Hao, et al. "VanillaKD... to large scale", arXiv 2023.
> [5] S Stanton, et al. "Does...really work?", NeurIPS 2021.
>
> **Finally, we will update the manuscript based on the above responses**. Regarding more experiments and discussions that we have made, you are referred to our responses to the other reviewers and in "Author Rebuttal by Authors".

---

> > ### Comment · Reviewer_mfZF · 2024-08-12
> >
> > Thanks for your response. Most of my concerns have been well addressed. I will maintain my initial score.

---

> > > ### Author Response · Authors · 2024-08-12
> > > **Thanks for the Recognition of Our Rebuttal**
> > >
> > > Thank you so much for the recognition of our responses. We are glad to see that you tend to accept our paper.
> > >
> > > We will make more efforts to improve our paper further.
> > >
> > > Many thanks for your constructive comments, time and patience.

---

> > ### Comment · Reviewer_3qQD · 2024-08-13
> >
> > Thanks for the author's response, I would like to increase my score

---

> > > ### Author Response · Authors · 2024-08-13
> > > **Thanks for the Recognition of Our Rebuttal**
> > >
> > > Thank you so much for the recognition of our responses. We are glad to see that you have raised your score from 6 to 7.
> > >
> > > We will make more efforts to improve our paper further.
> > >
> > > Many thanks for your constructive comments, time and patience.

---

### Official Review · Reviewer_rjnT · 2024-06-30

**Soundness:** 3
**Presentation:** 3
**Contribution:** 3
**Rating:** 6
**Confidence:** 5

**Summary:**

This paper presents a new knowledge distillation method, named ScaleKD. Previous works mostly use CNNs to distill vision transformers. However, how to use vision transformers to distill CNNs is less explored. This paper shows that pretrained vision transformers are good teachers for other types of student models. The cores of the proposed method are three main components, including the cross-attention projector, dual view feature mimicking, and teacher parameter perception.

The motivation of this paper is clear and the proposed method is also interesting. The authors propose to do model distillation from three aspects, which have been proven useful in the experiment section.

**Strengths:**

- The idea of this paper is interesting and the novelty is significant. Unlike previous KD methods, this paper proposes a new way to do knowledge distillation.

- The presentation of this paper is also good. It seems that the proposed method is easy to follow.

- Experimental results are good. Compared to previous knowledge distillation methods, the results shown in this paper improve them. In addition, the authors also provide detailed analysis on the importance of each component.

**Weaknesses:**

- Reading this paper is too tedious. The paragraph is too long in the introduction section. It is difficult to capture the important content.

- The proposed method consists of three parts, which make it look complicated.

- From Table 2, it seems that when the teacher models' scale increases, there is improvement. However, according to my knowledge, previous KD methods mostly fail to do this. Can the authors explain why the proposed method can achieve this? I think this is important for the KD community to design better KD methods.

- Though exploring how to use vision transformers as teachers is important, I am also curious about another thing. Have the authors used CNNs as teacher models? This is what most previous works did.

**Questions:**

I think the authors should elaborate more on why scaled teacher models help in the proposed method. I would like to see some analysis on this.

**Limitations:**

The authors have included limitations in the main paper.

---

> ### Author Rebuttal · Authors · 2024-08-06
>
> Thank you for the recognition of our work and the constructive comments.
>
> **1. To your concern regarding the presentation of the Introduction section,**
> >**Our response: (1)** Indeed, paragraphs in the Introduction section of our original manuscript are not short. Although the key messages are organized in a relatively dense manner, we think they are presented logically. In paragraph#1, we introduce the background on the evolution of network architectures and model learning paradigms and the problems. In paragraph#2, we first analyze the existing KD research and point out the weaknesses, then propose the motivation of our work, and finally elucidate three technical barriers for KD under our settings. In paragraph#3, we first describe the core ideas of our three components to address above barriers, then clarify how our method is formulated with the proposed components; **(2)** Following your suggestion, we will improve the presentation of the Introduction section, split long paragraph#2|#3 into two short paragraphs, making each paragraph be easy to capture the key content.
>
> **2. To your concern regarding the proposed method looks complicated,**
> >**Our response: (1)** Our method consists of three components, CAP, DFM and TPP, which are closely coupled to each other to address the differences in feature computing paradigm, model scale, and knowledge density. Our base component CAP is a neat cross-attention projector to transform CNN/MLP features into ViT-like features first, then DFM promotes feature alignment in both the original and frequency-aware feature spaces, inspired by two insightful observations (clarified in Line#98-105). Built upon CAP and DFM, TPP further establishes a proxy path to exploit the use of the pre-trained teacher's parameters to align the knowledge density discrepancy. In the ablation study, we validated the necessity of each component design (Table 9); **(2)** Thanks to the coupled design properties of CAP, DFM and TPP, the overall formulation of our method is simple, as discussed in Section 2.4.
>
> **3. To your comments regarding why our method could make pre-trained ViTs become good teachers,**
> >**Our response: (1)** Yes, many previous KD methods, such as [1-3], claim that large models are not always good teachers,  usually using CNNs as the teachers; **(2)** In the ViT era, ViT shows extraordinary scalability on model size and pre-training data size: i) larger model size provides ViT with stronger model learning capability; ii) the progress of pre-training makes large ViT learn better knowledge from more data. Thus, large ViTs are pre-trained on massive datasets, like IN-22K or LAION-2B. As a result, the capability gap between large-size models and small-size models in this era is larger than before, not only just suggested as the increasing model scale differences, but also as knowledge differences from pre-training data gaps; **(3)** Based on the above analysis, our work questions whether the student could inherit ViT teachers' scalable properties and underlines three important differences between teachers and students that lead to KD barriers. They are feature computing paradigm differences, model scale differences, and knowledge density differences. Accordingly, we design CAP, DFM and TPP to tackle the problems altogether. Fundamentally, we design CAP to directly align different computing paradigms across different architectures by employing patchfy stem and learnable queries. Progressively, we notice the other two types of differences are intertwined under the prevalent pre-training and fine-tuning paradigm, and are finally encoded in both teacher and student models’ feature space and parameter space. In light of this, DFM and TPP address the issues of model scale and knowledge density differences via considering feature distillation in the perspectives of feature space and parameter space. In feature space, DFM promotes feature alignment in both the original and frequency-aware feature spaces, giving more attention to subtle alternative components to promote feature space alignment. In parameter space, built upon CAP and DFM, TPP further establishes a proxy path to exploit the use of the pre-trained teacher's parameters to align the knowledge density discrepancy. As the three problems are well addressed, students can break down the barriers and inherit ViT teachers' capabilities. Thus, we can see the desired scalable performance trends in Table 2/3/6.
>
> >[1] S.I., Mirzadeh, et al. "Improved...teacher assistant", AAAI 2020.
> [2]  W. Son, et al. “Densely...multiple teacher assistants”, CVPR 2021.
> [3] T. Huang, et al. “Knowledge...stronger teacher”, NeurISP 2022.
>
> **4. To your comments on using CNN teachers,**
> >**Our response: (1)** We did not use CNNs as teacher models in the original manuscript since our work focuses on exploring the scalable properties of pre-trained ViT teachers to advance cross architecture KD research, as clarified in the Abstract and Introduction section; **(2)** To your request, with ResNet-50 as the student, we additionally use CovNeXt-XL as the teacher (adopted in the recent work [4]) besides Swin-L and BEiT-L. From the results shown below, we can observe that: i) ViT teachers show superiority to CNN teachers, e.g., ScaleKD achieves a notable 0.62% top-1 gain when the teacher is changed from ConNeXt-XL to BEiT-L; ii) Given ConNeXt-XL as the teacher, ScaleKD also outperforms [4] with a clear margin.
> |Teacher|Student|Method|Top-1(%)|
> |--|--|--|--|
> |ConNeXt-XL(86.97)|ResNet-50(79.80)|VanillaKD|81.10
> |ConNeXt-XL(86.97)|ResNet-50(78.64)|ScaleKD| 81.72
> |Swin-L(86.24)|ResNet-50(78.64)|ScaleKD|82.02
> |BEiT-L(88.58)|ResNet-50(78.64)|ScaleKD|82.34
>
> >[4] Z Hao, et al. "VanillaKD... to large scale", arXiv 2023.
>
> **Finally, we will update the manuscript based on the above responses**. Regarding more experiments and discussions that we have made, you are referred to our responses to the other reviewers and in "Author Rebuttal by Authors".

---

> > ### Comment · Reviewer_rjnT · 2024-08-10
> > **Final rating**
> >
> > Thanks for the reponses to my concerns. Basically, the authors have solved my concerns. In addition, all the other reviewers also recognize the contributions this paper made. I would like to keep my original rating unchanged.

---

> > > ### Author Response · Authors · 2024-08-10
> > > **Thanks for the Recognition of Our Rebuttal**
> > >
> > > Thank you so much for the recognition of our responses. We are glad to see that you tend to accept our paper.
> > >
> > > We will make more efforts to improve our paper further.
> > >
> > > Many thanks for your constructive comments, time and patience.

---

### Official Review · Reviewer_3qQD · 2024-07-05

**Soundness:** 3
**Presentation:** 3
**Contribution:** 3
**Rating:** 7
**Confidence:** 5

**Summary:**

This paper focuses on whether the pre-trained vision transformer models could be used as teachers to distilling knowledges to heterogeneous neural network architectures. The proposed ScaleKD aims to solve three problems including 1) feature computing paradigm different, 2) model scale differences, and 3) knowledge density differences. Extensive experiments are conducted with different neural networks, including CNN, ViT, and MLP on image classification task. This paper also shows that when scaling up the size of teacher models or their pre-training datasets, ScaleKD showcases larger gains to the student models.

**Strengths:**

1. The difficulties of transferring knowledge between different architectures are well summarized, including the differences in feature computing paradigm, differences in model scale, and differences in knowledge density.

2. All the figures, as well as the writing are clear and easy to follow.

3. The extensive experiments show the effectiveness of the proposed ScaleKD.

**Weaknesses:**

1. The related work section should be improved to discuss the differences between the proposed ScaleKD and existing works. The CAP, DFM, and TPP should be compared with existing works to show the originality and novelty. The references include:

[1] ViTKD: Feature-based Knowledge Distillation for Vision Transformers, CVPR 2024
[2] Prefallkd: Pre-impact fall detection via cnn-vit knowledge distillation, ICASSP 2023
[3] Distilling efficient vision transformers from cnns for semantic segmentation, Arxiv 2023
[4] A good student is cooperative and reliable: CNN-transformer collaborative learning for semantic segmentation, ICCV 2023

2. The experiments are conducted with ViT, SwinT, ResNet, MobileNet, ConvNeXt, MLP-Mixer. There are several types of CNN, however the claims in Abstract mentions that "Our method can train student backbones that span across a variety of CNN, MLP and ViT." More backbones should be included, or the description in abstract is over-claimed.

3. A question is the ScaleKD includes the MLP and ViT layers in CAP, does it mean the improved KD performance comes from such architecture update? Is the proposed method achieve KD with adding such layers to augment the student model? This is the main concern, please give more discussion.

**Questions:**

Please refer to the weakness.

---

> ### Author Rebuttal · Authors · 2024-08-06
>
> Thank you so much for the recognition of our work and the constructive comments.
>
> **1. To your comments about discussing more related works [1-4],**
> >**Our responses: (1)** Thanks for pointing out these four existing works which address KD between two transformers [1] or between the transformer and CNN [2-4]; **(2) Generally, our ScaleKD differs from these works in application focus, motivation, and method formulation. Firstly**, the application focuses of [2-4] are obviously different from ours. Our paper explores cross-architecture KD in the context of training backbone, fitting both classification and various downstream tasks, while [3-4] are KD methods specialized for semantic segmentation, and [2] is a KD application for fall detection (a task that aims to detect people's falls for avoiding injuries). **Secondly**, our motivation moves further from [1-4]. Specifically, they follow the training regime of traditional KD settings: i) the teachers are always small-to-medium-sized without pre-training (sharing the same dataset as the student); ii) the model size difference between teacher and student is not large. In sharp contrast, our ScaleKD explores how to help heterogeneous students approximate medium-to-large-size ViTs through new motivations: **how to connect KD research with state-of-the-art ViTs, aiming to help students mimic teachers' behaviors from the higher model capacity and the massive pre-training**, which shows clear differences to [1-4]; **Thirdly**, the formulation of ScaleKD is different from [1-4]. Basically, [1] uses distinct feature distillation strategies on shallow and deep layers, and [2] only conducts vanilla logits distillation between ViT teacher and CNN student. Differently, [3-4] have deeper insights and design methods specialized for semantic segmentation. To avoid heterogeneous features, [3] utilizes an MHSA layer to model global interdependencies on CNN features, and a channel-attention layer to build linguistic features, while [4] moves from offline KD to online collaborative learning and conducted feature distillation in early layers based on the similar mechanism as [3]. For model capacity gaps, [3] decouples the pixel-wise distillation by categories, while [4] employs a selective mechanism to ensure reliable distillation areas. In contrast, although our method also discusses related problems, the design ideas and method formulation have clear differences from [3,4]. In the context of learning from large pre-trained ViTs, our method underlines differences affecting the efficacy of KD from three aspects, feature computing paradigm, model scale and knowledge density. We first design CAP to tactfully align different computing paradigms across different architectures by employing patchfy stem and learnable queries. Progressively, we notice the other two types of differences are intertwined under the pre-training and fine-tuning paradigm, and are finally encoded in both teacher and student models’ feature space and parameter space. In light of this, DFM and TPP address the differences in model scale and knowledge density via considering feature distillation in the perspectives of feature space and parameter space.
>
> >[1] Z Yang, et al. "ViTKD...transformers", CVPR-W 2024.
> [2] TH Chi, et al. "Prefallkd...distillation", ICASSP 2023.
> [3] X Zheng, et al. "Distilling...segmentation", arXiv 2023.
> [4] J Zhu, et al. "A good student...segmentation", ICCV 2023.
>
> **2. To your comments on experiments with more ViT and MLP backbones,**
> >**Our response: (1)** Thank you for kind suggestion. Accordingly, we conducted extra experiments on more ViT and MLP students.  Due to the time limitation in the rebuttal phase, we performed experiments on two other typical models, ResMLP-S12 [5] and PVT-S [6], following the same setting as Table 3 in the manuscript.  From the results in the below table, we can see that both on ResMLP-S12 and PVT-S backbones, our method obtains promising top-1 gains, further verifying its good generalization ability to ViT and MLP backbones; **(2) Actually, the students in Table 3 are elaborately selected.** Specifically, we consider i) heterogeneous and homogeneous network pairs, ii) network pairs with different model capacity gaps, iii) the variety of the teacher and the student, and iv) the popularity of the networks. We initially selected 10 student models across 6 model types based on the above 4 principles. Now, we have 12 student models across 8 model types, which are more comprehensive. **The updated Table 3 is added to the PDF file attached in "Author Rebuttal by Authors"**.
> | Teacher|Student|Top-1(%)| $\Delta$ Top-1(%)|
> |--|--|:--:|:--:|
> |Swin-L(86.24)|ResMLP-S12(76.51)|80.54|+4.03
> |Swin-L(86.24)|PVT-S(79.80)|83.72|+3.92
>
> >[5] H Touvron, et al. "ResMLP...Training", TPAMI 2023.
> [6] W Wang, et al. "Pyramid ...Without Convolutions", ICCV 2021.
>
> **3. To your comments about whether the improvements come from the architecture updates,**
> >**Our response: (1)** Our ScaleKD does not alter the student network architecture as CAPs are connected to the student as auxiliary paths for feature distillation similar to conventional feature projectors. After the distillation training stage, all three components of ScaleKD, namely CAP, DFM and TPP will be removed, introducing no additional cost in the inference stage; **(2)** We guess it is Fig 1, especially Fig 1(c), that may result in this misunderstanding, where CAP seems to be involved in the student's network. For simplicity, in Fig 1, we only show the feature-mimicking process, not including the inference process; **(3)** Actually, we clarified this in Line#123-125 in the manuscript. We will revise Fig 1 and its caption to avoid this potential misunderstanding.
>
> **Finally, we will update the manuscript based on the above responses**. Regarding more experiments and discussions that we have made, you are referred to our responses to the other reviewers and in "Author Rebuttal by Authors".

---

> > ### Comment · Reviewer_3qQD · 2024-08-13
> >
> > I will set my final rate as "Accept" to support this paper

---

> > > ### Author Response · Authors · 2024-08-13
> > > **Thanks for Your Support of Our Paper**
> > >
> > > Thank you so much for recognizing our rebuttal and setting the final rate as "Accept" to support our paper.
> > >
> > > We will make more efforts to improve our paper further.
> > >
> > > Many thanks for your constructive comments, time and patience.

---

### Official Review · Reviewer_k9Ls · 2024-07-11

**Soundness:** 3
**Presentation:** 2
**Contribution:** 3
**Rating:** 6
**Confidence:** 4

**Summary:**

This paper concentrates on the distillation of knowledge from a large-scale, pre-trained, ViT-based teacher model to heterogeneous architectures. It incorporates three distinct designs: a) a Cross Attention Projector (CAP), which serves as the fundamental design that bridges the structural disparity between a non-ViT model and the ViT teacher; b) a Dual-View Feature Mimicking and a Teacher Parameter Perception module, both of which are constructed on top of the CAP to facilitate the distillation process. The effectiveness of the proposed methodology is validated through extensive experimentation.

**Strengths:**

- The proposed distillation technique is effective for heterogeneous architectures with a ViT-based teacher.
- The conducted experiments are comprehensive, providing solid validation for the effectiveness of the method.
- The proposed CAP structure is a good contribution which is feasible for various student architectures.

**Weaknesses:**

__Presentation__: The connection between DFM and TPP and their respective motivations is unclear. Specifically, it’s unclear how the modifications of DFM and TPP specifically address the issues of model scale and knowledge density.

__Effects of CAP__:  While the introduced CAP appears to be a promising design, its effectiveness is only demonstrated within the context of the ScaleKD framework. It would be interesting to see if this architecture could be integrated with other heterogeneous architecture methods, such as OFA, to replace traditional projectors.

__Effects of DFM__: The ablation study of DFM merely indicates that the filtering operation is beneficial to CAP-based distillation. It raises the question of whether this strategy is also compatible with other feature alignment architectures, such as linear head or convolutional head?

__Effects of TPP__: The design of TPP seems to be at odds with its stated purpose. Given that different network architectures, such as CNNs or ViTs, are known to have different knowledge preferences, it’s questionable whether aligning the student’s parameters with the teacher’s under a heterogeneous architecture is an effective approach. Furthermore, it’s unclear why using the last layer of the teacher as an additional head function for distillation would aid in alignment in the parameter space. Empirical evidence (such as CKA visualization) or theoretical justification is needed to demonstrate whether TPP can encourage different network architectures to exhibit similar learning behavior.

__Experiments__:  Some details of the experiments, such as the teacher information in Tables 6-8, are missing from the presented tables.

**Questions:**

please refer to the weakness part.

**Limitations:**

The limitations are properly discussed.

---

> ### Author Rebuttal · Authors · 2024-08-06
>
> Thank you so much for the recognition of our work and the constructive comments.
>
> **1. To your comments about the presentation of DFM and TPP,**
> > **Our responses: (1)** The basic motivation of our DFM and TPP components is to align model scale and knowledge density differences in a joint but not separate manner, **under the premise that**: our basic component CAP (a novel cross-attention projector containing positional embedding, patchify stem and trainable queries) has already aligned feature computing paradigm differences between the ViT teacher and the CNN/MLP/heterogeneous ViT student and made the student have the same tokens-based feature modeling in terms of semantic units and spatial resolution as the teacher to perform feature distillation; **(2)** According to our analysis (Line#68-81), model scale differences make teacher and student models have different learning capacities, and knowledge density differences are mainly due to the pre-training data which is supposed to be visible only for the teacher in our work. As a result, these differences are intertwined under the prevalent pre-training and fine-tuning paradigm and are finally encoded in both teacher and student models’ feature space and parameter space. In light of this, DFM and TPP address the issues of model scale and knowledge density differences via considering feature distillation in the perspectives of feature space and parameter space;  **(3)** In principle, i) DFM relies on an insightful observation that shows the frequency feature distributions of pre-trained ViTs are extremely imbalanced (dominated by the direct component). Inspired by this, DFM uses a novel dual-view feature mimicking formulation to promote feature alignment in the original and frequency-aware feature spaces; ii) TPP relies on another critical observation that the pre-training knowledge is only gained by the ViT teacher as the pre-training data is invisible to the student. Based on this, TPP forms a cross-network proxy parameter alignment path via bridging the early stages (projected by CAP) of the student to the later stages of the ViT teacher, exploiting the use of the pre-trained teacher's parameters to further reduce the knowledge density discrepancy; **(4)** Generally, CAP, DFM and TPP are progressively designed, and they are closely connected to each other as discussed in (Line#51-127, Section 2) and validated in Table 9/13, Fig 2/3/5/6.
>
> **2. To your comments about the effects of CAP to other heterogenous KD methods,**
> >**Our responses:** Following your suggestion, we added an ablation by applying CAP to OFA. According to the results shown below, our CAP brings 0.27% top-1 gain to OFA with traditional projectors, validating its superiority.
> |Teacher|Student|Method|Top-1(%)|
> |--|--|--|:--:|
> |DeiT-T(72.17)|ResNet-18(69.85)|OFA|71.33
> |DeiT-T(72.17)|ResNet-18(69.85)|OFA+CAP|71.60
>
> **3. To your comments about the effects of DFM on other feature projectors,**
> >**Our response: (1)** Yes, our DFM would be compatible with existing feature projectors such as Linear and Conv projectors, as it stems from the observation of the teacher's feature distribution; **(2)** To study this compatibility, we added an ablation, following your suggestion. From the results shown below, we can see: i) DFM brings 0.35%|0.45% top-1 gain to Linear|Conv projector, validating the compatibility of DFM with them; ii) Comparatively, the top-1 gain of DFM to CAP is much higher, indicating the coupled design property of DFM and CAP.
> |Method|Top-1(%)|$\Delta$ Top-1(%)|
> |--|:--:|:--:|
> Baseline|76.55|-|
> Linear|77.43|+0.88
> Linear+DFM|77.78|+1.23
> Conv|77.52|+0.97
> Conv+DFM|77.97|+1.42
> CAP|77.87|+1.32
> CAP+DFM|78.51|+1.96
>
> **4. To your comments about the effects of TPP,**
> > **Our responses: (1)** We first hope this concern is alleviated after you read our responses to your first concern. Given heterogeneous teacher and student network architectures, it is indeed not reasonable to use TPP directly, as connecting the student’s early-stage parameters with the teacher’s later-stage parameters seems to be a conflict due to different feature computing paradigms (lead to different knowledge preferences); **(2)** However, in our method, TPP is built upon CAP which well aligns feature computing paradigm differences and DFM which jointly considers feature distributions in the original and frequency-aware feature spaces, paving a strong base to exploit the use of the pre-trained teacher's parameters to reduce the knowledge density discrepancy. For simplicity, in implementation, we use the last stage of the teacher to construct the proxy parameter alignment path, which already attains promising performance (Table 9) yet it may be not optimal (e.g., using the last two stages of the teacher is slightly better, bringing ~0.15% extra top-1 gain); **(3)** Following your suggestion, we add CKA visualizations to illustrate the effects of TPP.  As shown in Fig 1 of the PDF file in "Author Rebuttal by Authors", ScaleKD encourages the student to have similar behaviors as the teacher at the last stage where TPP is applied.
>
> **5. To your comments about teachers' information to experiments in Tables 6-8,**
> >**Our responses: (1)** There are no teachers for the experiments on downstream tasks in Table 7-8. As described in Line#187-189, the experiments were performed based on the student model trained by ScaleKD on IN-1K to verify whether the performance gain could be well preserved to downstream tasks in the standard transfer learning regime; **(2)** In Table 6, when comparing ScaleKD with supervised and self-supervised methods, our teacher is Swin-L pre-trained on IN-22K. When comparing ScaleKD with the other two pre-training paradigms, our teacher is BEiT-L pre-trained by EVA.
>
> **Finally, we will update the manuscript based on the above responses**. Regarding more experiments and discussions that we have made, you are referred to our responses to the other reviewers and in "Author Rebuttal by Authors".

---

> > ### Comment · Reviewer_k9Ls · 2024-08-10
> > **Post rebuttal comment**
> >
> > Thanks for the response, which addresses my concerns. I will increase my rating accordingly. However, I still think the statement of 'parameter space' is somewhat ambiguous. I suggest the authors include more clarification in the final version.

---

> ### Author Response · Authors · 2024-08-10
> **Thanks for the Recognition of Our Rebuttal**
>
> Thank you so much for the recognition of our responses. We are glad to see that you have raised your score.
>
> We will improve the statement and clarification related to 'parameter space' regarding our TPP component and continue to make more efforts to improve our paper further.
>
> Many thanks for your constructive comments, time, and patience.

---

### Author Rebuttal · Authors · 2024-08-06

Dear Reviewers, Area Chairs, Senior Area Chairs and Program Chairs,

We sincerely thank all four reviewers for their thorough and constructive comments. We are glad that the novelty, method component designs, validation pipeline and performance of our work have been mostly recognized by all four reviewers.

In the past week, we carefully improved the experiments (using all computational resources we have), the clarifications and the discussions of our work to address the concerns, the questions and the requests by all four reviewers. **Summarily, we made the following improvements:**
> (1)  To have a better understanding of specific component designs in our ScaleKD framework, we follow the constructive comments/requests by Reviewer k9Ls and Reviewer mfZF, and add several ablation studies and analytical experiments, including: i) An ablation study on the compatibility of DFM with other types of feature alignment architectures; ii) An ablation study on verifying the necessity of non-zero frequency features in the first mimicking path of DFM; iii) An ablation study on the compatibility of CAP with other cross-architecture KD methods; iv) Analytical experiments on how TPP affects the student's learning behaviors in our ScaleKD framework.
(2) To further strengthen the motivation of our work, we follow the constructive comments/requests by Reviewer 3qQD, Reviewer rjnT, and Reviewer mfZF, and provide: i) Experiments that compare ViT teachers and CNN teachers; ii) Experiments on more teacher-student network pairs; iii) Detailed analysis on why ScaleKD shows scalability in terms of the pre-trained ViT teacher's capability; iv) Comparisons with more related works.
(3) We also provide detailed responses to the other concerns/questions/requests raised by each reviewer one by one.

Finally, in the attached one-page PDF file, all the aforementioned experimental results are summarized in different Tables. We will include the above experiments and discussions in our final paper. We hope our detailed responses are helpful to address the concerns, the questions and the requests of all four reviewers.

---

### Decision · Program_Chairs · 2024-09-25

**Decision:**

Accept (poster)

**Comment:**

All reviewers agree on the quality of the paper that provide a technically solid distillation method with experiments to support is effectivity. I recommend acceptance for this paper.